

# Spatialize v1.0: A Python/C++ Library for Ensemble Spatial Interpolation

Felipe Navarro [1,2], Alvaro F. Egaña [1,2], Alejandro Ehrenfeld [1,2], Felipe Garrido [1,2], María Jesús Valenzuela [1,2], and Juan F. Sánchez-Pérez [3]

[1]Advanced Mining Technology Center (AMTC), Universidad de Chile, Avenida Tupper 2007, RM, Chile, URL: http://www.amtc.cl
[2]Advanced Laboratory for Geostatistical Supercomputing (ALGES), Department of Mining Engineering - AMTC, Faculty of Mathematical and Physical Sciences, Universidad de Chile, Avenida Tupper 2069, RM, Chile, URL: http://www.alges.cl
[3]Department of Applied Physics and Naval Technology, Universidad Politécnica de Cartagena (UPCT), Cartagena, Spain

**Correspondence:** Felipe Navarro (felipe.navarro@amtc.cl)

**Abstract.** In this paper, we present Spatialize, an open-source library that implements *ensemble spatial interpolation*, a novel method that combines the simplicity of basic interpolation methods with the power of classical geostatistical tools, like Kriging. It leverages the richness of stochastic modelling and ensemble learning, making it robust, scalable and suitable for large datasets. In addition, Spatialize provides a powerful framework for uncertainty quantification, offering both point estimates and empirical posterior distributions. It is implemented in Python 3.x, with a C++ core for improved performance, and is designed to be easy to use, requiring minimal user intervention. This library aims to bridge the gap between expert and non-expert users of geostatistics by providing automated tools that rival traditional geostatistical methods. Here, we present a detailed description of Spatialize along with a wealth of examples of its use.

## 1 Introduction

A significant challenge in the field of geosciences is the issue of sparsity that is often observed in spatial databases, such as soil properties, climate data, or mineral concentrations, which are characterised at limited point locations. (Li and Heap, 2014). The presence of these data gaps hinders a comprehensive understanding of the variable's domain. The central issue, therefore, is estimating values at unmeasured locations. Various spatial interpolation algorithms have been developed for this purpose.

Geostatistics is a field focused on the analysis, estimation, and modelling of spatial variables. Unlike traditional statistics, geostatistics emphasises the spatial dependencies between observations (Maroufpoor et al., 2020). The Kriging technique, the most relevant exponent of geostatistical interpolation (McKinley and Atkinson, 2020; Kirkwood et al., 2022b), was initially devised for the estimation of gold reserves (Kleijnen, 2017; Virdee and Kottegoda, 1984). As an unbiased linear estimator that minimises estimation error at each position, Kriging is commonly referred to as BLUE (Best Linear Unbiased Estimator) (McKinley and Atkinson, 2020; Fischer and Getis, 2009; Varouchakis et al., 2012; Abzalov, 2016). Beyond providing robust estimates, Kriging also facilitates the calculation of estimation variance, which is widely used for assessing spatial prediction uncertainty (Abzalov, 2016; Varouchakis et al., 2012). A variety of user-friendly tools are available for the implementation



of Kriging, including PyKrige, PySAL, gstat, automap, geoR, and fields. Nevertheless, it should be noted that the use of Kriging without actual knowledge of the model may result in suboptimal and misleading outcomes (Oliver and Webster, 2014; Assibey-Bonsu, 2017). In particular, parameter selection and spatial continuity modelling have a significant effect on

the accuracy of Kriging estimates (Abzalov, 2016; Chilès and Desassis, 2018; Pannecoucke et al., 2020). However, the correct determination of these inputs requires substantial expertise and data (Fischer and Getis, 2009; Wang et al., 2017; Pannecoucke et al., 2020; de Sousa Mendes et al., 2020), which creates a significant barrier for most potential users. Moreover, the task becomes increasingly complex when the variables under study are of a dynamic, spatio-temporal nature (Samson and Deutsch, 2022; Boroh et al., 2022), or are not structurable as a regular grid (Oliver and Webster, 2014).

In summary, spatial interpolation tasks, when assessed from the perspective of classical geostatistical analysis, can be time-consuming and require considerable expertise. Consequently, there is a need for more straightforward yet effective spatial interpolation methods that can address highly dynamic spatial problems without necessitating manual spatial analysis tasks.

In contrast to geostatistical approaches, deterministic models employ straightforward calculations; nevertheless, they are only capable of producing estimations (Li and Heap, 2014). The most widely applied of these methods is inverse distance

weighting (IDW), a simple yet powerful spatial interpolation method that uses a weighted average of surrounding point values to estimate the unknown value at an unsampled location (Mitáš and Mitášová, 1988). In recent years, variants of IDW have been successfully used in a variety of applications, including estimation of air pollution levels (LI Jialin, 2017), soil moisture (Abdulwadood et al., 2021) and water quality (Khouni et al., 2021). The main limitation of IDW is that it does not take into account the spatial structure or correlation of the variable being interpolated. This can lead to over-smoothing or under-

smoothing of the estimated values, depending on the degree of spatial correlation in the data (Li, 2021).

Another promising approach for spatial interpolation is the use of machine learning-based methods, which can learn complex spatial relationships from large datasets without requiring manual spatial modelling (Li et al., 2011; Kirkwood et al., 2022a). For example, Leirvik and Yuan (2021) and Wang et al. (2019) proposed deep learning-based spatial interpolation methods to estimate solar radiation and interpolate seismic data, respectively. Nevertheless, challenges that arise from deep learning

models are, firstly, the need for large amounts of data and computational resources to train them, and secondly, the necessity to measure additional variables other than the one under study. This is especially true for complex spatial-temporal problems, where the number of input variables and temporal observations can be substantial (Hamdi et al., 2022). A further challenge in using deep learning for spatial interpolation is the difficulty of interpreting the results. These models are often referred to as 'black boxes', meaning that the process by which predictions are derived remains uncertain. This can be problematic in

situations where transparency and interpretability are important, such as in environmental applications (Paudel et al., 2023; Qingmin Meng and Borders, 2013; Susanto et al., 2016).

In addressing the need for a simple and flexible spatial interpolation technique, able to adapt to highly dynamic phenomena, scalable to big data, interpretable, and most importantly widely accessible to the entire geoscientific community, Menafoglio et al. (2018b) and Egaña et al. (2021) independently proposed a new state-of-the-art spatial interpolation method based on

ensemble learning. This method combines the simplicity of methods such as IDW with the power of Kriging spatial analysis, which the authors of the latter named Ensemble Spatial Interpolation (ESI). This model is able to provide reliable estimates





that are comparable to those of Kriging, while eliminating the need for manual spatial continuity modelling. Its main features are: (a) it is based on a stochastic space partitioning process, which aids in managing large datasets; (b) it is built under an ensemble scheme, which guarantees robustness despite the use of weak local interpolation functions with small subsets of data,

and (c) it provides a powerful framework for uncertainty quantification, as it is based on a Bayesian scheme, thus yielding an empirical posterior distribution of the estimate (instead of a single point estimate).

The aim of this article is to present `spatialize`, a novel software library that facilitates an efficient implementation of ESI. `Spatialize` has been designed to be easy to use, efficient and flexible. The core of the library is implemented in C++ with a Python 3.x programming API. It is available as an open-source project, making it accessible to researchers and

practitioners in industry and academia. The subsequent sections provide a comprehensive overview of the ESI model and the Spatialize library, including its features and capabilities. We also present several examples of how the library can be used in practical applications. Finally, the future directions of the library and its potential impact on spatial interpolation research and practice will be discussed.

## 2    Ensemble spatial interpolation

Ensemble learning is usually regarded as the statistical and computational conception of the "wisdom of the crowd", whose idea is to collect and combine the points of view of many experts to produce an ensemble result (Egaña et al., 2021). An ensemble model $\hat{z}$ can be formulated as:

$$\hat{z} = G(f_1(x^*), \cdots, f_m(x^*)) \tag{1}$$

Where $x^*$ is a vector of covariances and $\{f_1, \cdots, f_m\}$ is a set of weak voter (regression, classification, interpolation, etc.) functions. Function $G$ is an aggregation function that combines the responses from each voter function and it can be as simple as majority voting for classification, (Friedman et al., 2000; Collins et al., 2002; Džeroski and Ženko, 2004; Hothorn and Lausen, 2005; Reid and Grudic, 2009), averaging for regression, or more sophisticated approaches such as a mixture of experts (MoE) (Jacobs et al., 1991; Jacobs, 1995; Jordan and Xu, 1995; Cohen and Intrator, 2000).

## 80    2.1    Weak voter function set generation

In the ESI model, the construction of the set of weak voter functions is achieved through the concept of "bootstrapping the space" (Egaña et al., 2021), which involves generating different spatial configurations through random partitioning of the space where the data are located. These partitions are generated in a way that any combination of data is possible, while preserving their spatial locations and avoiding data clustering (Figure 1). Each partition creates unique data subsets within the

partition cells, where any spatial interpolation method can be applied. An unmeasured location is then estimated by combining (aggregating) the estimates derived from all the partition elements across the set of partitions where that location falls (Egaña et al., 2021).





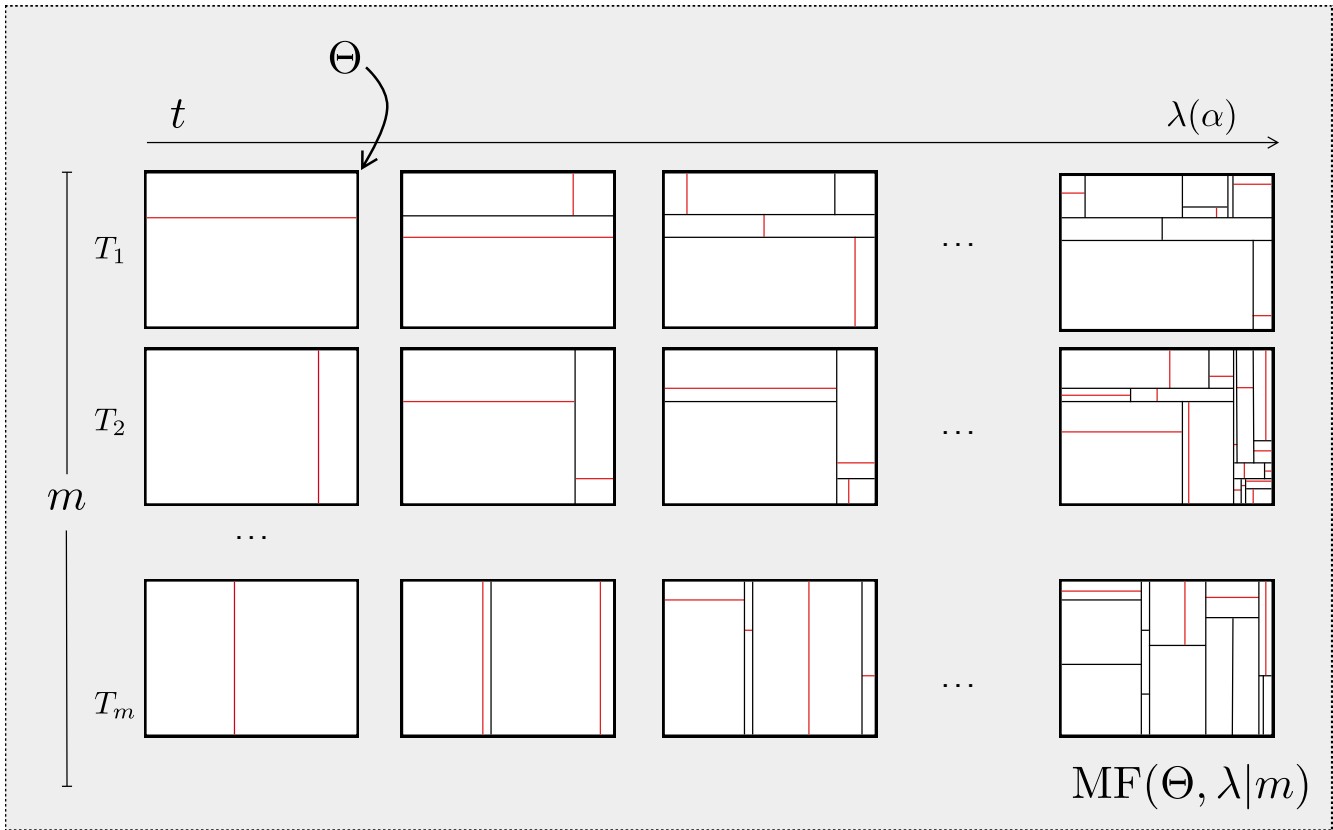

**Figure 1.** Generative process to draw a time-dependent stochastic partition set of size $m$. Time goes along the x-axis, ending at time given by $\lambda(\alpha)$. Red lines indicate the current time partition cut. *Note:* Reprinted from "Ensemble Spatial Interpolation: A New Approach to Natural or Anthropogenic Variable Assessment", by Egaña et al. (2021), *Natural Resources Research*, *(30)*, 3777–3793.

It is well known that a spatial partition data structure can be represented as a tree (Samet, 1984), where nodes represent partition spaces and edges indicate containment relationships. This representation enables efficient spatial data querying and operations on the data contained in the spaces, similar to how certain related hierarchical data structures, such as k-d trees and octrees, are used in spatial indexing. Thus, in practice, generating a set of random partitions is equivalent to generating a forest of random tree structures, analogous to how multiple decision trees form a random forest.

The Spatialize library uses two methodologies to generate these partitions of space:

**a) Mondrian Forests (MF):** As proposed by Egaña et al. (2021) and introduced by Lakshminarayanan et al. (2014). The latter is a non-parametric Bayesian strategy that is employed for both classification and online regression, which has been demonstrated to be as efficacious as other high-level ensemble learning methods, such as random forests (Breiman, 2001) or additive trees (Hastie et al., 2009).





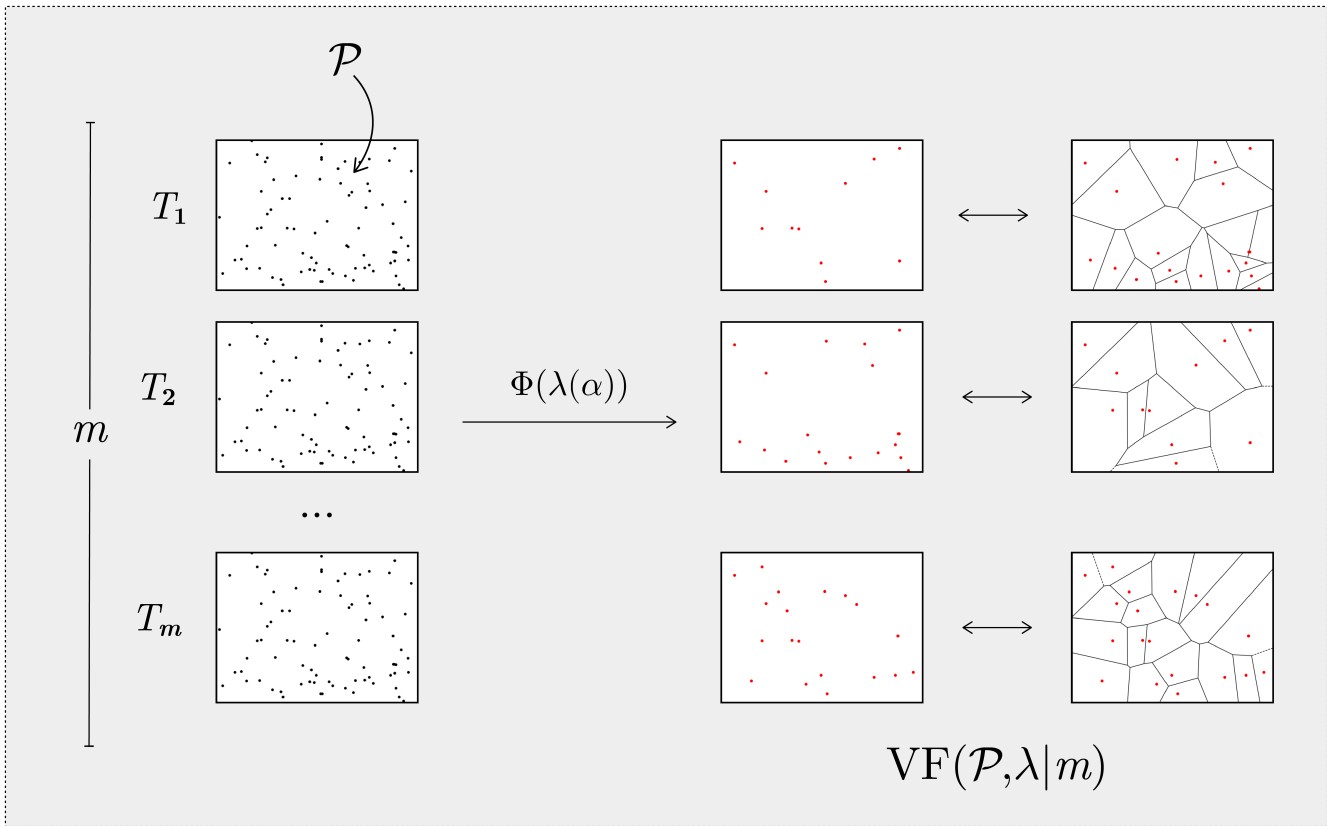

**Figure 2.** Generative process to draw a topological stochastic partition set of size $m$. The number of Voronoi nuclei is given by $\lambda(\alpha)$. Red points indicate the selected locations, among the sampled locations, for the nuclei.

In the context of ESI, MF is a collection of random tree structures defined as:

$$\text{MF}(\Theta, \lambda | m) = \{T_1, \cdots, T_m\}, \ T_k \sim \text{MT}(\Theta, \lambda) \tag{2}$$

Where $m$ is the number of random tree structures in the forest and $\text{MT}(\Theta, \lambda)$ is the generative process, described in Algorithm 1, which produces random samples of tree structures. The latter represents random space partitions of the target domain $\Theta$ that are assumed to be $\Theta = [a_1, b_1] \times \cdots \times [a_d, b_d] \subset \mathbb{R}^d$.

Algorithm 1 implements a temporal stochastic process that generates nested random partitions aligned with coordinate axes. Parameter $\lambda$ represents the process's finite lifetime, while $\tau$ represents the elapsed time through recursion levels. The recursive partitioning process is as follows: Line 8 samples the 'time' until the next cut in the sub-box $\theta$ from an exponential distribution, parametrized such that $\mathbb{E}(E) = 1/\mu(\theta)$, where $\mu(\theta) = \sum_{i=1}^{d}(b_i^\theta - a_i^\theta)$. This ensures smaller sub-boxes are less likely to be partitioned. In line 10, the dimension to be partitioned is selected, where $p_k$ determines the probability of selecting dimension $k$. For a sub-box $\theta$, $p_k$ is proportional to $(b_k^\theta - a_k^\theta)$, favouring the partition of larger sides. In line 11, the cut point is randomly



---

**Algorithm 1** $\mathrm{MT}(\Theta, \lambda)$ Sampling Algorithm

---

1: **procedure** Main()
2: SampleMT$(\Theta, \lambda)$ $\{\Theta = [a_1, b_1] \times \cdots \times [a_d, b_d]\}$
3: **end procedure**
4: **procedure** SampleMT$(\theta, \lambda)$
5: SampleMTBranch$(\theta, \lambda, 0)$
6: **end procedure**
7: **procedure** SampleMTBranch$(\theta, \lambda, \tau)$
8: $E \sim \mathrm{Exp}(\mu(\theta))$ $\{\mu$ is a measure on $\mathbb{R}^d\}$
9: **if** $(\tau + E) < \lambda$ **then**
10:    $d_x \sim \mathrm{Discrete}(p_1, \ldots, p_d)$
11:    $x \sim U([a_{d_x}, b_{d_x}])$
12:    SampleMTBranch$(\theta^>, \lambda, (\tau + E))$
13:    SampleMTBranch$(\theta^<, \lambda, (\tau + E))$
14: **end if**
15: **end procedure**

---

determined along the selected dimension, creating sub-boxes $\theta^>$ and $\theta^<$. Figure 1 illustrates the partitioning process, with each red line representing a cut at time $t$ for all $T_k$.

   **b) Voronoi Forests (VF):** A variation of a) which employs Voronoi partitions instead of Mondrian trees, in a manner analogous to that described by Menafoglio et al. (2018a). However, rather than using a fixed number of nuclei, Spatialize employs a random number per partition, calibrated to ensure that the expected number of data points per cell matches that of a
Mondrian tree.

   Thus, a Voronoi Forest (VF) is a collection of structures defined as:

$$\mathrm{VF}(\Theta, \lambda | m) = \{T_1, \cdots, T_m\}, \quad T_k \sim \mathrm{VT}(\Theta, \lambda) \tag{3}$$

   where $\mathrm{VT}(\Theta, \lambda)$ is the generative process that produces a Voronoi partition. The process begins with the selection of $K$, the number of Voronoi nuclei, which is drawn from a Poisson distribution with parameter $\lambda$. Next, a random sample of $K$
nuclei, denoted by $\Phi_K = \{c_1, ..., c_K\} \subseteq \Theta$, is randomly generated. Finally, the target domain $\Theta$ is partitioned by assigning all contained locations $x \in \Theta$ to the nearest nuclei based on the Euclidean distance. A Voronoi cell is thus defined by $\mathcal{L}_i = \{x \in \Theta : \|x - c_i\| \leq \|x - c_j\| \, \forall c_j \in \Phi_K, j \neq i\}$. The process of generating a Voronoi Forest is illustrated in Figure 2.





### 2.1.1 Model training

Let us define *conditioning data* as a set $\mathcal{M} = \{z_j\}_{j=1}^{N_s}$ of $N_s$ measurements of a variable of interest, obtained at specific spatial

locations $\mathcal{P} = \{\mathbf{x}_j\}_{j=1}^{N_s}$ within a particular region of a $d$-dimensional space. The classical formulation of spatial interpolation can be stated as: find a $d$-variate function $\mathbf{S}_{(\mathcal{P},\mathcal{M})}$ that fulfils the condition $\mathbf{S}_{(\mathcal{P},\mathcal{M})}(\mathbf{x}_j) = z_j$, $j = \{1, \cdots, N_s\}$ [1].

Now, let us assume $\mathcal{P} \subset \Theta$. Then, both a Mondrian Forest, $\mathrm{MF}(\Theta, \lambda | m)$, and a Voronoi Forest, $\mathrm{VF}(\Theta, \lambda | m)$, can be 'trained' when a set of $d$-dimensional data points (the conditioning data) are used to condition the sampling of $\{T_1, \cdots, T_m\}$. We see, then, that:


#### a) When using Mondrian Forests

A trained Mondrian Forest is defined as:

$$\mathrm{MF}(\Theta, \lambda | \mathcal{P}, m) = \{T_1, \cdots, T_m\}, \ T_k \sim \mathrm{MT}(\Theta, \lambda | \mathcal{P}) \tag{4}$$

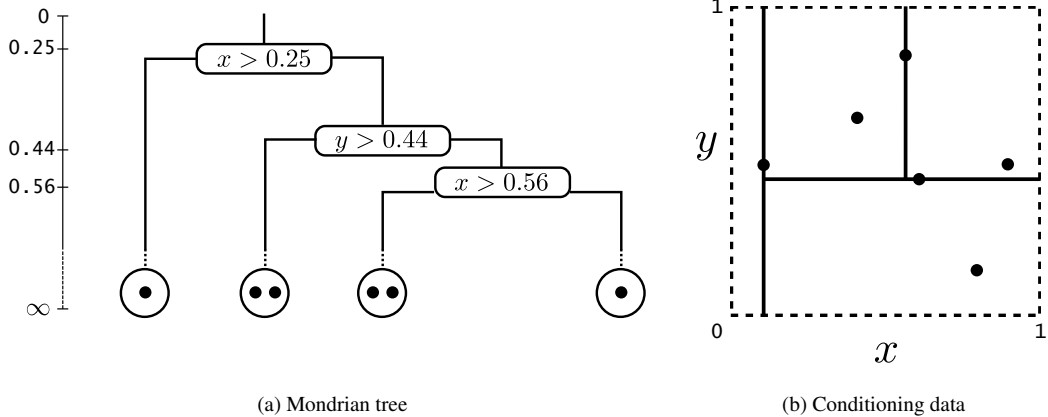

(a) Mondrian tree          (b) Conditioning data

**Figure 3.** Tree structure schema. (a) decision tree structure. (b) The spatial partition corresponding to the decision tree structure shown in (a). Black points represent conditioning data. *Note:* Reprinted from "Ensemble Spatial Interpolation: A New Approach to Natural or Anthropogenic Variable Assessment", by Egaña et al. (2021), *Natural Resources Research*, *(30)*, 3777–3793.

A Mondrian tree can be trained by conditioning the partitioning process to the data (Figure 3b). Thus, a trained random

partition set, $\mathrm{MF}(\Theta, \lambda | \mathcal{P}, m)$, can be obtained by modifying Algorithm 1 as follows:

– For any box $\theta$ define $\theta^* = k(\theta, \mathcal{P})$ as the smallest sub-box containing all conditioning positions in $\theta$ (see Figure 4).

– The probability of splitting a sub-box (line 8) is replaced by: Sample $E \sim Exp(\mu(\theta^*))$.

– Lines 10 and 11 in Algorithm 1 are adjusted to work on $\theta^*$ instead of $\theta$.

---

[1]The sub-index $(\mathcal{P}, \mathcal{M})$ indicates that the interpolation function is constructed using both the values of the measurements and their locations.





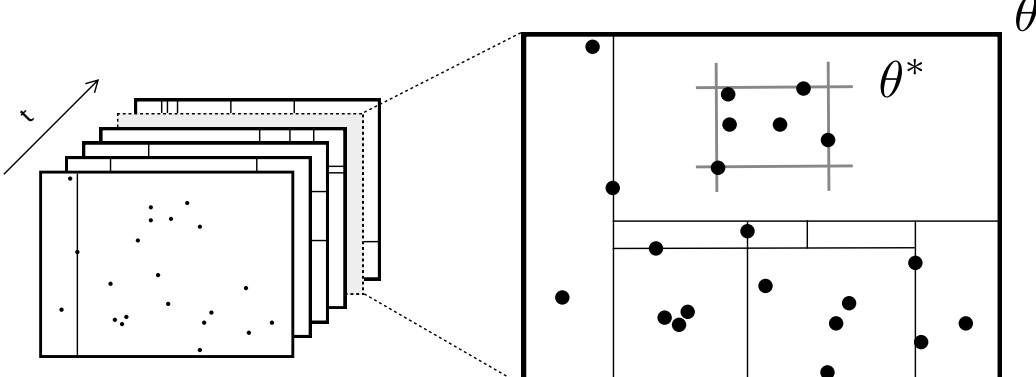

**Figure 4.** Illustration of $\theta^*$ given a set of data points, which enables Algorithm 1 to be trained using sample data. *Note:* Reprinted from "Ensemble Spatial Interpolation: A New Approach to Natural or Anthropogenic Variable Assessment", by Egaña et al. (2021), *Natural Resources Research*, *(30)*, 3777–3793.

As a result of this modification, sub-boxes containing highly concentrated data are more likely to be partitioned. This offers

the advantage of ensuring that most leaf nodes (i.e. the resulting sub-boxes) contain a reasonable amount of conditioning data, thus avoiding spatial clustering.

**b) When using Voronoi Forests**

A trained Voronoi Forest (VF) is defined as:

$$\mathrm{VF}(\Theta, \lambda | \mathcal{P}, m) = \{T_1, \cdots, T_m\}, \quad T_k \sim \mathrm{VT}(\Theta, \lambda | \mathcal{P}) \tag{5}$$

A trained random partition set, $\mathrm{VF}(\Theta, \lambda | \mathcal{P}, m)$, can be obtained by sampling the $m$ sets of Voronoi nuclei from the locations at which measurements of the variable of interest are available, as opposed to being sampled from $\Theta$. For each tree, this results in the set $\Phi_K = \{c_1, \cdots, c_K\}$ where $K \leq N_s$ and $\Phi_K \subseteq \mathcal{P}$.

Thus, the process to obtain samples from $\mathrm{VT}(\Theta, \lambda | \mathcal{P})$ is as follows:

– Sample $K \sim Poisson(\lambda), 1 \leq K \leq N_s$.

– Sample $\Phi_K = \{c_1, ..., c_K\}$ from the measured locations $\mathcal{P}$

– Establish each Voronoi cell as $\mathcal{L}_i = \{x \in \Theta : \|x - c_i\| \leq \|x - c_j\| \; \forall c_j \in \Phi_K, j \neq i\}$.

Sampling from the measured locations $\mathcal{P}$ ensures that all partition cells $\mathcal{L}_1, \cdots, \mathcal{L}_K$, generated by sampling $\mathrm{VT}(\Theta, \lambda | \mathcal{P})$,

will contain at least one measured location.



### 2.1.2 Weak voter function set

For an unmeasured position $x^* \in \Theta$, we define $\mathcal{L}_k \subset (\mathcal{P}, \mathcal{M})$ as the set of conditioning data points contained within the partition cell where $x^*$ falls into in tree $T_k$.

Then, let us consider a base interpolation function $\mathbf{S}_{(\mathcal{P}, \mathcal{M})}$, which can be any spatial interpolator for which no additional information, other than measurements and their locations, is required to interpolate new positions – such as Kriging or IDW.

Now, for each tree $T_k$, let $\mathbf{S}_{\mathcal{L}_k}$ be the base interpolation function restricted to $\mathcal{L}_k$. Thus, the $k^{th}$ weak voter function for $x^*$ is obtained by applying the base interpolator $\mathbf{S}_{(\mathcal{P}, \mathcal{M})}$ to estimate the value at $x^*$ using only the points in $\mathcal{L}_k$.

Formally, the weak voter function set is defined as:

$$f_k(x^*) = \mathbf{S}_{\mathcal{L}_k}(x^*), \quad k = 1, \cdots, m \tag{6}$$

## 2.2 Interpolation using the trained model

Let us denote $x_k^* = f_k(x^*) = \mathbf{S}_{\mathcal{L}_k}(x^*)$ as the $k^{th}$ weak voter function for $x^*$. Then, $\{x_k^*\}_{k=1}^m$ corresponds to the set of weak voter functions for $x^*$ resulting from all $m$ trees.

Thus, the interpolation function $\mathcal{Z}_{(\mathcal{P}, \mathcal{M})}$ corresponds to the aggregation of these weak voter functions:

$$e^* = \mathcal{Z}_{(\mathcal{P}, \mathcal{M})}(x^*) = G(\{x_k^*\}_{k=1}^m) \tag{7}$$

The simplest choice for the aggregation function $G$ is the mean $\mathbb{E}[\cdot]$. In this case, the interpolation function becomes:

$$e_{\mathbb{E}}^* = \frac{1}{m} \sum_{k=1}^m x_k^* \tag{8}$$

## 2.3 Interpolation precision modelling

A precision model $p^*$ for $\mathcal{Z}_{(\mathcal{P}, \mathcal{M})}(x^*)$ can be defined using a loss function $\mathbb{L}$ as follows:

$$p^* = \mathbb{E}_{\hat{P}(\{x_k^*\}_m)}(\mathbb{L}(e^*, \{x_k^*\}_m)) \tag{9}$$

Equation 9 represents a generalisation of the mean-variance concept within the context of Bayesian variability.

When using the mean-based interpolator $e_{\mathbb{E}}^*$, we can define an associated interpolation variance (or error) $\mathbb{V}_{e_{\mathbb{E}}^*}(x^*)$ as:

$$p_{\mathbb{E}}^* = \mathbb{V}_{e_{\mathbb{E}}^*}(x^*) = \frac{1}{m} \sum_{k=1}^m (x_k^* - e_{\mathbb{E}}^*)^2 \tag{10}$$





## 2.4 Rule of thumb for parameter choice

### 2.4.1 When using Mondrian Forests

The domain parameter $\Theta$ can be considered as any bounding box containing the positions of the conditioning data $\mathcal{P}$. Thus, in practice, the only two parameters of the model are:

– The number of partitions (or tree structures) $m$. A reasonable suggestion for this parameter is that higher is better, keeping in mind that higher values will directly impact time performance. Experiments have shown that certain stability is reached for $m \geq 500$ (Egaña et al., 2021), so this would be a good starting point.

– The process lifetime $\lambda$. The only restriction for this parameter is that it must be positive. This renders the selection, or any sensitivity analysis, of its value challenging. In order to address this issue, a function of a normalised parameter $\alpha \in [0, 1)$ is used to obtain suitable values for $\lambda$, which is defined as follows:

$$\lambda(\alpha) = \frac{1}{\mu(\Theta)(1 - \alpha)} \tag{11}$$

In practice, $\alpha$ controls the average tree depth in the forest, determining how finely the space is partitioned. In this way,
$\alpha = 0$ will generate the coarsest partition, while $\alpha \to 1$ will generate finer ones. $\alpha$ must be carefully chosen to ensure that the base interpolation function $\mathbf{S}_{\mathcal{L}_k}$ has sufficient sample data. Once $m$ has been defined, it is recommended to use cross-validation to find the optimal $\alpha$, typically within $[0.7, 0.95]$.

### 2.4.2 When using Voronoi Forests

As seen in Mondrian Forest, the domain parameter $\Theta$ may be regarded as any bounding box encompassing $\mathcal{P}$. Consequently,
the parameters of the model remain identical, yet their respective roles and practical considerations differ:

– The number of partitions (or tree structures) $m$. A reasonable suggestion for this parameter is that higher is better, keeping in mind that higher values will directly impact time performance. However, the possibility of reaching stability for a certain value of $m$ is yet to be studied in the context of Voronoi Forests.

– The process lifetime $\lambda$. This parameter determines the expected number of Voronoi nuclei for each partition. In the
context of Voronoi Forest, $\lambda$ is related to the mean of the Poisson distribution. Although the only theoretical restriction for this parameter is that it must be greater than one, in Spatialize, it is also constrained to match the expected number of leaves in a Mondrian tree. To this end, $\lambda$ is calculated by multiplying the parameter $\alpha$, defined in the context of Mondrian forests, by a factor according to the number of observations $N_s$, as follows:

$$\lambda(\alpha) = \frac{1}{2} * N_s * \alpha, \alpha \in [0, 1) \tag{12}$$

As in the case of Mondrian partitions, $\alpha$ also controls how coarse or fine the partitions are by affecting the number of Voronoi nuclei. Thus, $\alpha = 0.25$ will generate the coarsest partition, while $\alpha \to 1$ will generate finer ones.





In the context of sampling from a trained tree $\text{VT}(\Theta, \lambda | \mathcal{P})$, additional considerations arise with respect to $m$ and $\lambda$. Firstly, the number of nuclei, $K(\lambda(\alpha))$, must not exceed the number of observations, $N_s$. Secondly, as $K(\lambda(\alpha)) \to N_s$, when sampling from $\mathcal{P}$, the number of possible Voronoi nuclei combinations. Therefore, a large $m$ value may result in duplicated partitions. To avoid such inefficiencies, it is recommended that $m \ll C(N_s, K)$, or alternatively, not employing trained trees (also possible on Spatialize). Alternatively, Spatialize also permits the direct sampling of $\Phi_K$ from $\Theta$ (i.e. not employing trained trees).

## 3 Usage examples

The main motivation behind the development of the `spatialize` library was to provide the scientific and technical community with a robust and automated spatial estimation tool that can be used across different disciplines by researchers and professionals who are not experts in geostatistics.

In this section, we present a sequence of usage examples which illustrate how `spatialize` facilitates the task of automated spatial estimation, including hyper-parameter search functions and estimation functions, as described in detail in the User Manual (supplementary material).

Furthermore, these examples demonstrate that the efficacy of our tool is comparable to or superior to that of other automatic spatial estimation tools, taking into account parameters obtained through automated grid searches – a process that does not necessitate the involvement of an expert to select a priori. To make the point, examples of estimation generated with the `SciPy` library are also presented.

### 3.1 Gridded data estimation

Estimation with ESI on data that is on a regular grid is performed with the `esi_griddata()` function. Two examples have been developed and are described below.

Firstly, we present a use case in which the reference two-dimensional surface is a cubic-type function. The idea of this example is to compare `spatialize` with various types of spatial estimation within the `SciPy` library, using a discrete sampling of a cubic-type function, while also illustrating the estimation process with gridded data using the `spatialize` library. Then, a comparative analysis of the two partitioning methods (Mondrian and Voronoi) is presented. Furthermore, we introduce the `esi_hparams_search` function, which searches for optimal ESI parameters, and the `loss` functionality, which allows for the definition of custom loss functions.

The Code snippet 1 shows how to generate the grid and some points and the corresponding values generated by the cubic type function – taken from the documentation of the `griddata` function of the module `scipy.interpolate`, included in the `SciPy` library. This dataset is used as input for all examples performing estimations where data are included in a regular grid. Figure 5 illustrates this function alongside randomly sampled points, which are to be used in the comparison of the different interpolation methods.





---

**Code snippet 1** Generation of the points and values that are input to the spatial gridded estimation examples presented below.

```
def func(x, y):  # a kind of "cubic" function
return x * (1 - x) * np.cos(4 * np.pi * x) *
np.sin(4 * np.pi * y ** 2) ** 2
grid_x, grid_y = np.mgrid[0:1:100j, 0:1:200j]
rng = np.random.default_rng()
points = rng.random((1000, 2))
values = func(points[:,0], points[:,1])
```

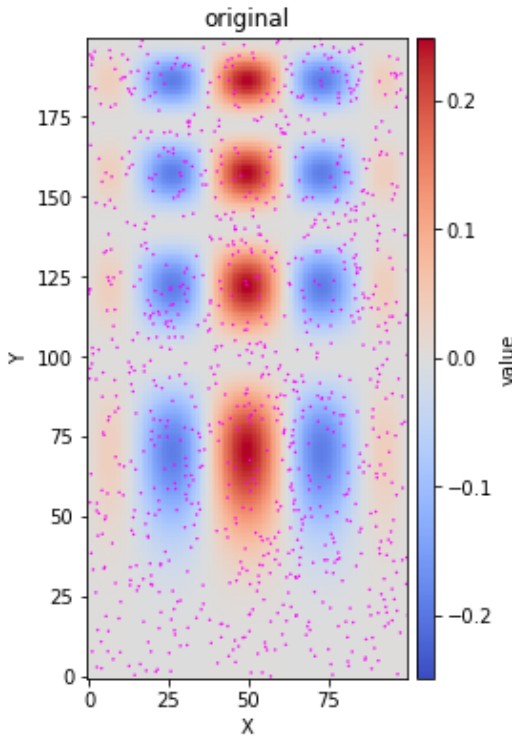

**Figure 5.** The original cubic type function, plotted in superposition to the randomly generated points from which the spatial interpolations will be made.

### 3.1.1 ESI vs scipy

As mentioned above, in order to have a comparison with the results of known interpolation tools implemented in Python, we use the `griddata` function of the `scipy.interpolate` module with the 'nearest neighbour', 'linear' and 'cubic' methods. Code snippet 2 shows how to estimate with this function the three cases that are shown in Figure 6.

245



**Code snippet 2** Griddata function to generate three different estimations using SciPy library.

```
from scipy.interpolate import griddata
nearest_result=griddata(points, values, (grid_x, grid_y), method='nearest')
linear_result=griddata(points, values, (grid_x, grid_y), method='linear')
cubic_result=griddata(points, values, (grid_x, grid_y), method='cubic')
```

Then, we use the `esi_griddata` estimation function with the local IDW and Kriging interpolators and arbitrary non-optimal parameter sets. Code snippets 3 and 4 show the two implementations.

**Code snippet 3** Gridded estimation using IDW as local interpolator.

```
esi_griddata(points, values, (grid_x, grid_y),
             local_interpolator="idw",
             p_process="mondrian",
             data_cond=False,
             exponent=1.0,
             n_partitions=500, alpha=0.985,
             agg_function=af.mean)
```

**Code snippet 4** Gridded estimation using Kriging as local interpolator.

```
esi_griddata(points, values, (grid_x, grid_y),
             local_interpolator="kriging",
             model="spherical", nugget=0.0, range=10.0,
             n_partitions=500, alpha=0.9,
             agg_function=af.mean)
```

In Figure 6, we can see the original shape generated with the cubic function and the estimations made with ESI, based on IDW and Kriging as local interpolators and the three interpolation options generated with the SciPy library. It can be observed that the results produced by ESI, without parameter optimisation and without any structural assumption, are quite acceptable compared to those generated by the SciPy interpolators, which have a structural assumption defined a priori. Obviously, if one introduces an inductive bias by somehow knowing that the underlying function to be interpolated is cubic, the cubic model will have the best result, at least visually. We can call this the inductive bias effect.

250





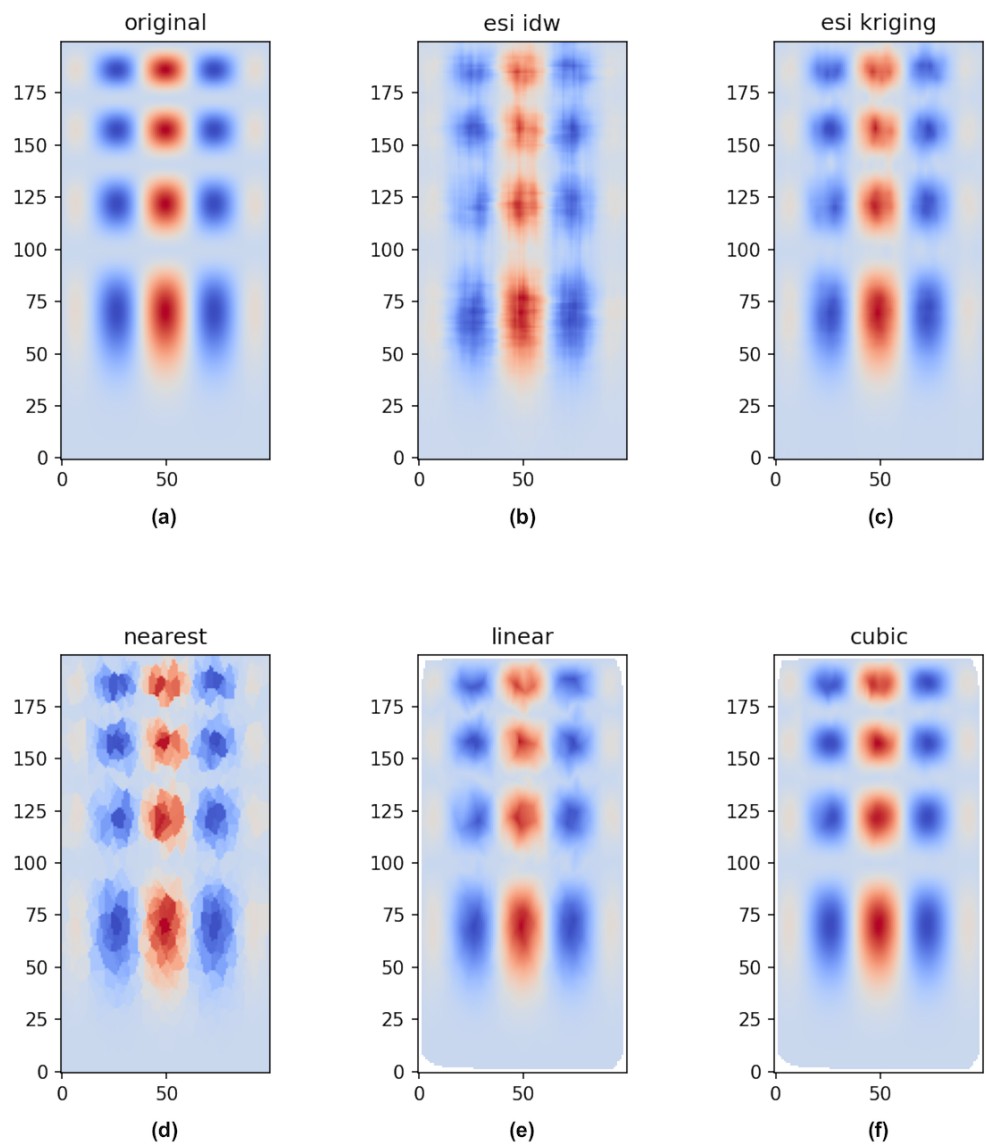

**Figure 6.** Comparison of ESI and SciPy estimates for the same gridded data set. a) The original cubic type function; b) ESI IDW interpolation; c) ESI Kriging interpolation; d) SciPy nearest-neighbour interpolation; e) SciPy linear interpolation; f) SciPy piecewise cubic interpolation.





As mentioned within Section 2.1, both Mondrian Forest and Voronoi Forest can be used as the partitioning method in the
case of ESI-IDW[2]. Code snippet 5 shows how to generate both versions of estimation, in the gridded case, with the IDW
interpolator.

---

**Code snippet 5** Gridded estimation using IDW as local interpolator and using both Mondrian and Voronoi partition methods.

```
esi_griddata(points, values, (grid_x, grid_y),
local_interpolator="idw",
p_process="mondrian",
data_cond=False,
exponent=1.0,
n_partitions=500, alpha=0.985,
agg_function=af.mean
)
esi_griddata(points, values, (grid_x, grid_y),
local_interpolator="idw",
p_process="voronoi",
data_cond=True, # and False alternatively
exponent=1.0,
n_partitions=500, alpha=0.985,
agg_function=af.mean
)
```

---

Figure 7 shows both ESI-IDW partitioning alternatives: Mondrian Forest and Voronoi Forest. In the latter case, the case
without and with data conditioning is shown. It can be seen that activating data conditioning makes the estimation look more
like the case where Mondrian Forest is used for partitioning, while not using data conditioning makes the estimation look
smoother, i.e., closer to the original scenario. This situation looks interesting because it reflects the fact that the partitioning
process can significantly affect the outcome. Surely, there must be some Bayesian generative argument behind this, which
clearly deserves more attention.

---

[2]For the case of the local kriging interpolator, `spatialize` only has the Mondrian Forest implementation.



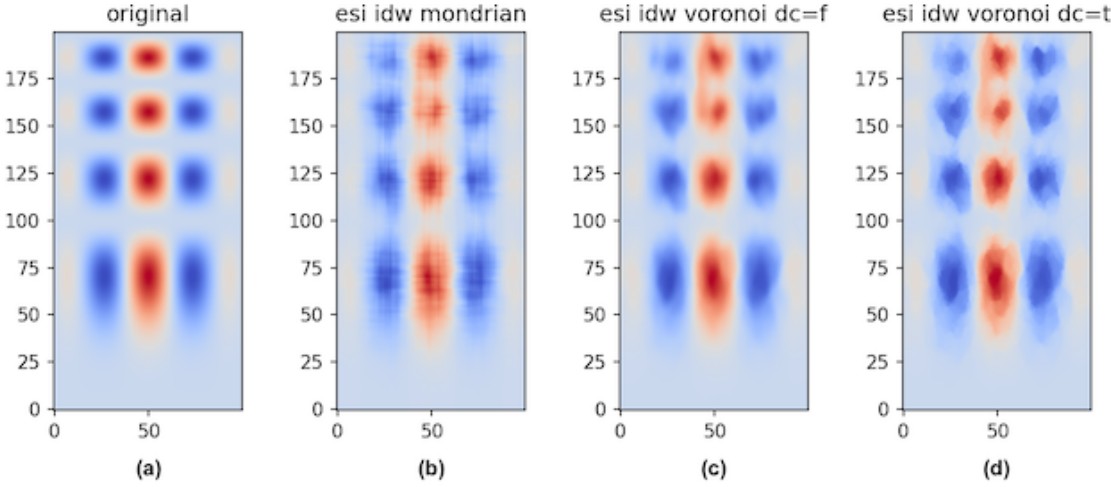

**Figure 7.** Comparison of ESI-IDW estimates for Mondrian and Voronoi partition methods, for the two graphs on the right, $dc = data\_cond$
represents whether there is (t) or not (f) conditioning on the data. a) The original cubic type function; b) ESI IDW interpolation using
Mondrian partition; c) ESI IDW interpolation using Voronoi partition without conditioning on the data; d) ESI IDW interpolation using
Voronoi partition with conditioning on the data.

### 3.1.2 Hyper parameter search

One of the powerful elements of the `spatialize` library is the facility to automate the search for the best parameters for

ESI estimation. This search is performed by the function `esi_hparams_search`, which is available for both the gridded
and ungridded estimation cases. This function employs cross-validation to determine the parameter combination that yields the
minimum error out of a previously defined set.

The purpose of this section is to provide an example of the use of the grid search functions, focusing on the gridded case. In
this example, we first conduct a search from a set of options for each of the parameters, and then perform the estimation with

the best parameters found. We then compare the results with the best estimate that can be achieved using the IDW estimator as
a global estimator without ESI, to give a reference for the improvement implied by using it only as a local interpolator in ESI.
In the case of IDW without ESI, the `spatialize` library includes a grid search function, which is analogous to the one for
ESI estimation parameters search.

A parameter search is now performed on the same two-dimensional data set that was previously used, employing the

`esi_hparams_search` function. As can be seen in Code snippet 6, the search function receives ranges or sets of arguments, which specify different combinations of parameters for interpolation. In the example, Kriging is used as the local




interpolator, with four different omnidirectional variogram model options. The function generates and compares all possible scenarios with the combinations of these parameters.

**Code snippet 6** Grid search to find the best case parameters within the given ranges and options for optional named arguments in `esi_hparams_search` function, in this case for the Kriging local interpolator.

```
search_result =
esi_hparams_search(points, values, (grid_x, grid_y),
local_interpolator="kriging", griddata=True, k=10,
model=["spherical", "exponential", "cubic", "gaussian"],
nugget=[0.0, 0.5, 1.0],
range=[10.0, 50.0, 100.0, 200.0],
alpha=[0.97, 0.96, 0.95])
```

Figure 8 shows the error frequency graphs (left) and the different error levels for each scenario (right). In Code snippet 6,
the object `search_result` will contain the result data for the grid search executed. It is interesting to observe the result in this graphical way, where one can see how the level of estimation error is distributed (histogram on the left) and then how this error evolves in the sequence of scenarios running during the search.

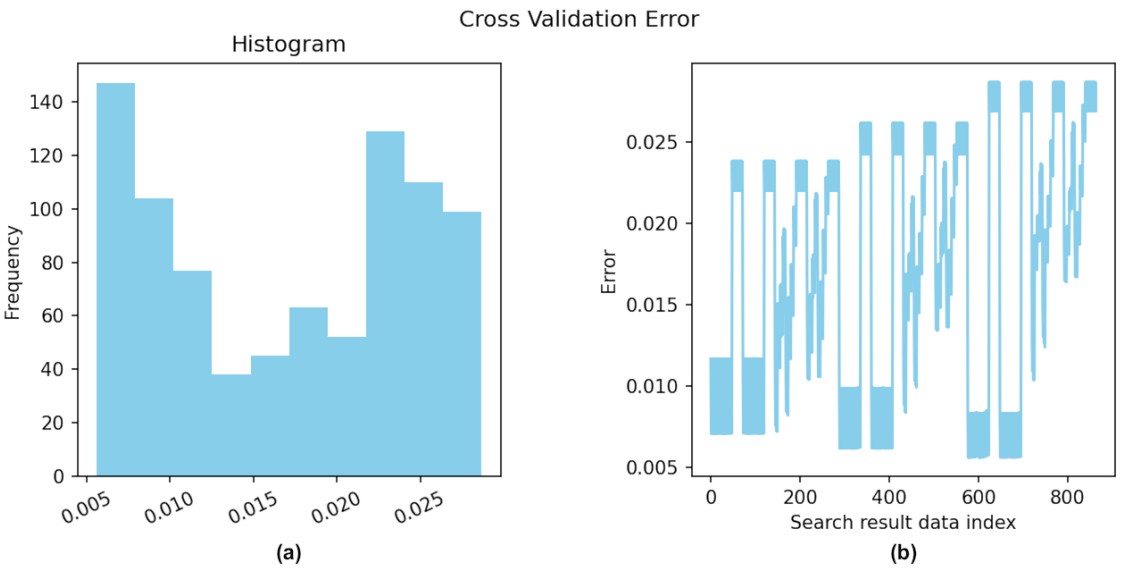

**Figure 8.** Cross-validation error for grid search using Kriging as local interpolator. a) Histogram of errors; b) Errors in the sequence of scenarios during the search.

Finally, the estimation is performed based on the parameters of the scenario with the lowest cross-validation error, which in this case has an index of 302. Since this is an interpolation, and therefore, the estimate at points where there is grid data
is by construction equal to the reference data, to calculate the error, $K$ points are removed in a K-Fold round and the error of



the estimate at those points is calculated, and then the average of all iterations is obtained. In this example, $K = 10$. It can be noted in the code below that the gridded estimation function only requires as optional arguments those provided by the method `search_result.best_result()`.

---

**Code snippet 7** Gridded estimation using best result of `esi_hparams_search` result.

```
1  result = esi_griddata(points, values, (grid_x, grid_y),
2                        best_params_found=search_result.best_result()
3                        )
```

---

Figure 9 shows the result of the estimation with the best parameters obtained from the previous search (left). The model that
delivered the search is the 'spherical' one, with an $alpha = 0.95$. It is remarkable how the structure of the original image is recovered, considering that there are no domains or variographic studies involved in this model. On the right is the accuracy, calculated with the default loss function (MSE). It can be seen that the largest impressions appear around the boundaries of the structures.

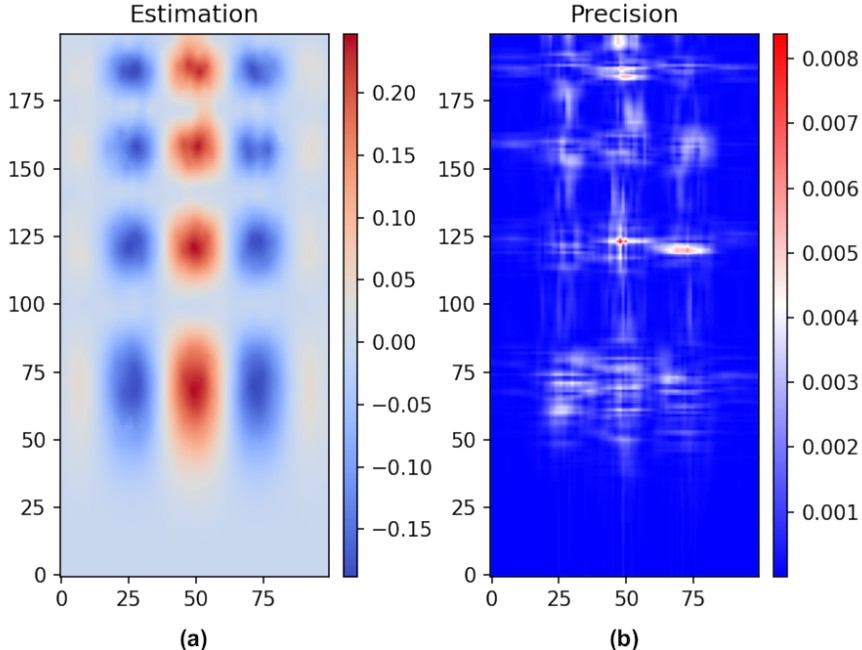

**Figure 9.** a) ESI grid search best estimation using Kriging as local interpolator; b) Precision obtained with MSE (default) loss function.

Now, to show how powerful this spatial interpolation method is, we compare the best-case estimations of ESI with IDW
as the base interpolator and an interpolation that only uses IDW. The aim is to show that this method does have the ability to rescue the structural aspects of the presented example, provided that they are regular shapes (cubic type function, in this case).





First, we search for optimal IDW hyperparameters using the `idw_hparams_search` function, analogous to the `esi_hparams_search` function. Code snippet 8 shows its implementation.

**Code snippet 8** Grid search for IDW estimation (without ESI).

```
search_result =
idw_hparams_search(points, values, (grid_x, grid_y),
griddata=True, k=10,
radius=[0.07, 0.08],
exponent=(0.001, 0.01, 0.1, 1, 2)
)
```

As shown in Figure 10, the minimum errors obtained are in the order of 0.016.

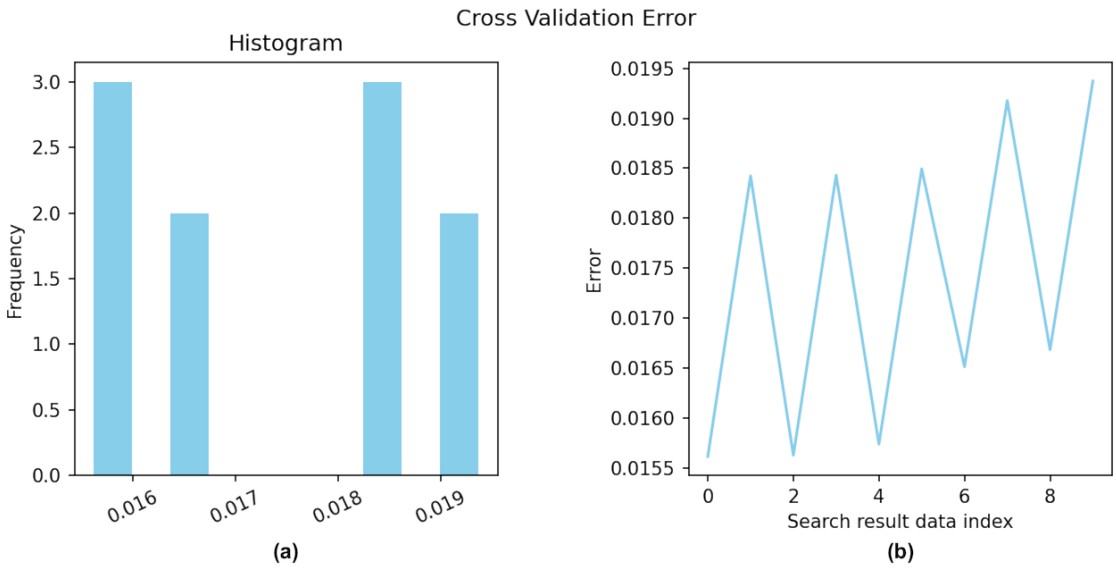

**Figure 10.** Cross-validation error for grid search for IDW interpolator without ESI. a) Histogram of errors; b) Errors in the sequence of scenarios during the search.

We can then generate the best estimate from the best hyperparameters found in the search, as shown in code snippet 9.

**Code snippet 9** Gridded estimation with IDW without ESI, using IDW grid search best result.

```
1  result =
2  idw_griddata(points, values, (grid_x, grid_y),
3               best_params_found= search_result.best_result(optimize_data_usage=False))
```




Figure 11 shows the best possible estimation for the search parameter grid performed in the pure IDW case. It can be seen that the IDW interpolator does rescue the original image structures.

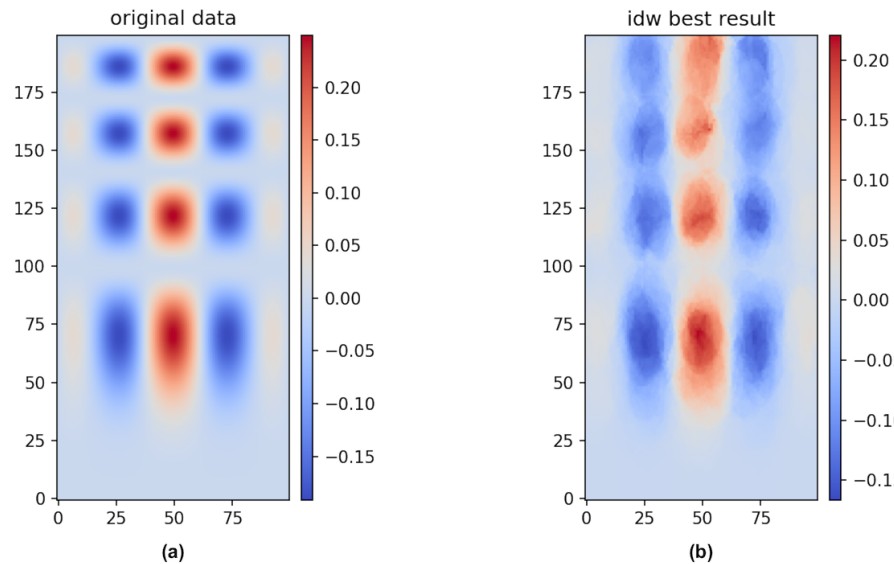

**Figure 11.** Best estimation using IDW interpolator without ESI. a) The original cubic type function; b) Traditional IDW interpolation.

Now, we will use IDW as a local interpolator for ESI, performing the same parameter search and estimation exercise. The code shown in Code snippet 10 allows us to do this operation. For the parameter search, we have defined a wide set of

305 combinations, where we even consider different aggregation functions to be applied on the sets of ESI scenes obtained in each case. In addition, the Voronoi partitioning method has been used since it generates more regular partition elements, which may be favourable in this case, given that the local IDW interpolator works radially.




---

**Code snippet 10** Grid search to find the best case parameters within the given ranges and options for optional named arguments in `esi_hparams_search` function, in this case for IDW local interpolator with gridded data ESI estimation.

---

```
search_result =
esi_hparams_search(points, values, (grid_x, grid_y),
local_interpolator="idw", griddata=True, k=10,
p_process="voronoi",
n_partitions=(30, 50, 100),
exponent=[0.001, 0.01, 0.1, 1, 2],
alpha=(0.95, 0.97, 0.98, 0.985),
agg_function={"mean": af.mean,
"median": af.median,
"p25": af.Percentile(25),
"p75": af.Percentile(75)
})
```

---

Figure 12 shows the errors for the 400+ scenarios generated by the search. The minimum errors obtained are in the order of 0.012. In this case, the power of ESI is expressed in 25% lower error levels than in the pure IDW case (Figure 10).

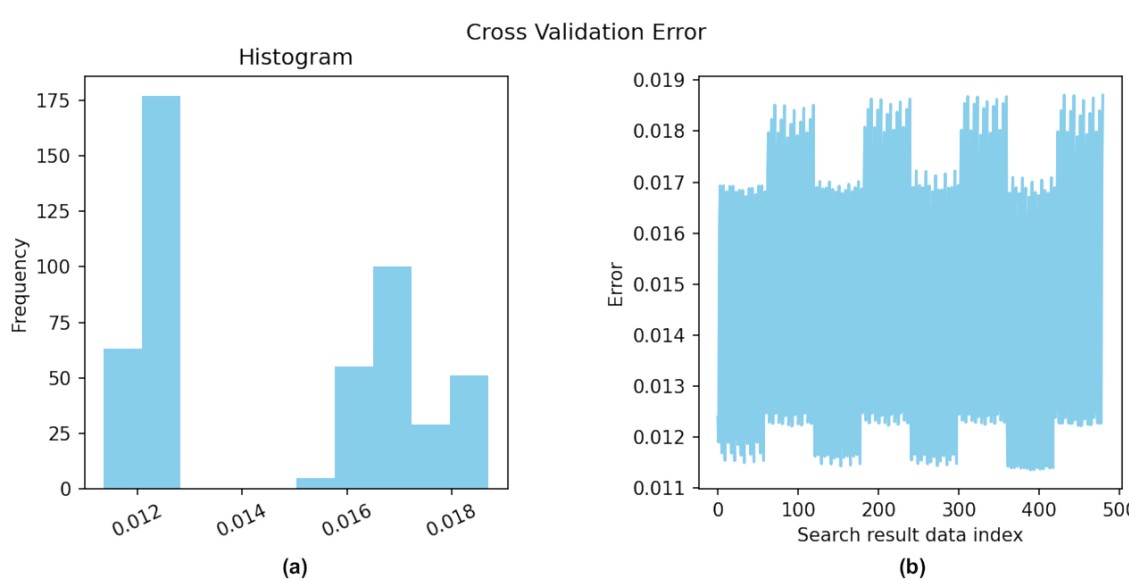

**Figure 12.** Cross-validation error for grid search using IDW as local interpolator.a) Histogram of errors; b) Errors in the sequence of scenarios during the search.

Finally, analogous to the case of ESI-Kriging estimation, code snippet 11 performs the ESI-IDW estimation with the parameters found in the search.





---

**Code snippet 11** Gridded estimation with IDW using grid search best result.

```
1  result = esi_griddata(points, values, (grid_x, grid_y),
2                        best_params_found=search_result.best_result()
3                        )
```

---

Finally, Figure 13 presents the estimation with the best parameters found by the grid search above. Although the accuracy in this case is a little lower than in the case of the ESI-Kriging estimation, this result is noticeably better than the one presented with the basic IDW interpolation. It can be seen that capturing the structures of the original function is a result of the power of
ESI.

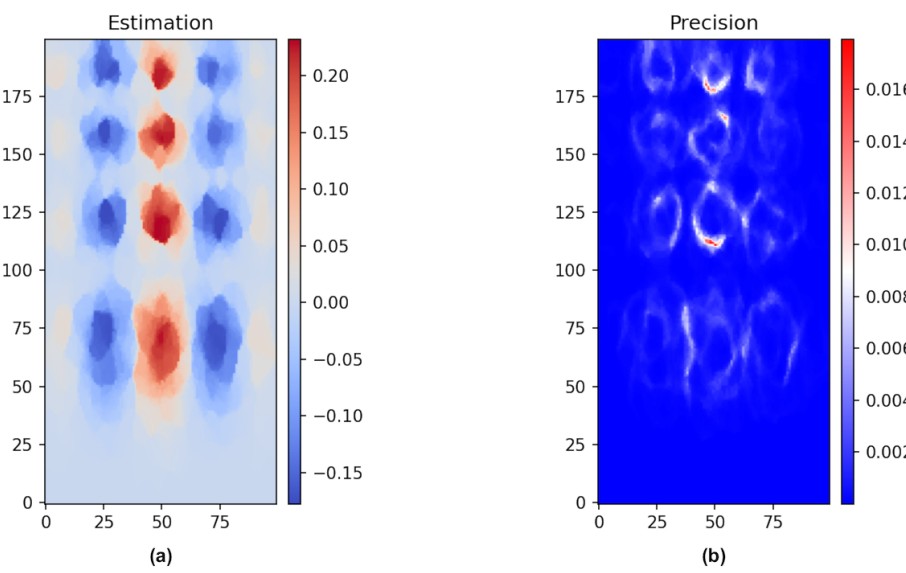

**Figure 13.** a) ESI grid search best estimation using IDW as local interpolator; b) Precision obtained with the Operational Error function.

### 3.1.3 Custom precision functions

In this section, we will review the process of implementing accuracy metrics through relationships other than mean-variance.

If we would like to implement our own loss function for the calculation of custom precision from the scenarios generated by the ESI estimator and a particular aggregate estimate, `spatialize` provides a very modular and convenient way to do
so. For example, to have our own implementation of the operational error, which is also implemented as a class in the library (module `lossfunction`), we would use the following code contained in the example mentioned above.





---

**Code snippet 12** Example of how to create a customised loss function, in this case, a version of the operational error.

```python
from spatialize.gs.esi.lossfunction import loss

def op_error_precision(estimation, esi_samples):
    dyn_range = np.abs(np.nanmin(esi_samples) - np.nanmax(esi_samples))

    @loss(af.mean)
    def _op_error(x, y):
        return np.abs(x - y) / dyn_range

    return _op_error(estimation, esi_samples)
```

---

As can be seen in the above code, the function is defined, and within it, a second function is defined, which is decorated by `@loss()`, within which is also assigned the aggregation function that will be used to aggregate the unit loss calculations per scenario into a single precision layer. Figures 14 and 15 show a comparison between the default accuracy contained in the `ESIResult` object and that produced by our operational error function for the ESI-IDW (in this case, using Mondrian partitions) and ESI-Kriging cases, respectively.

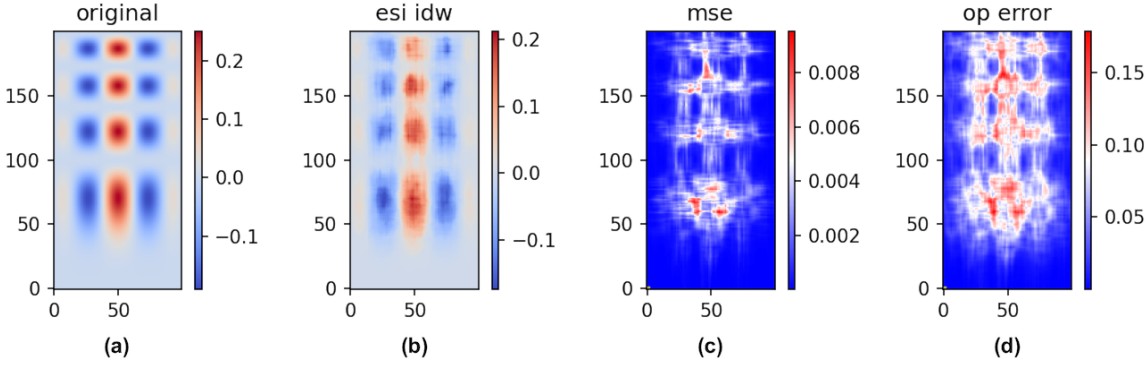

**Figure 14.** Mean square error and operational error for an ESI-IDW interpolation. a) The original cubic type function; b) ESI IDW interpolation using Mondrian partition; c) Mean-squared error; d) Operational error.





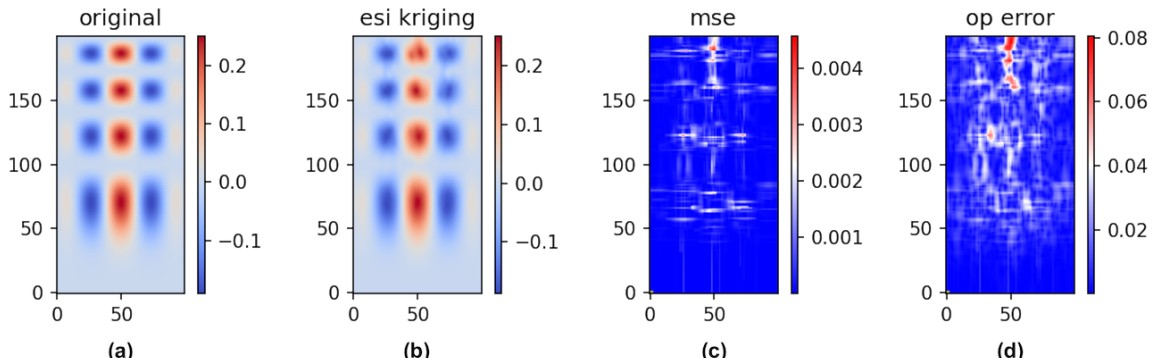

**Figure 15.** Mean square error and operational error for an ESI-Kriging interpolation. a) The original cubic type function; b) ESI Kriging interpolation using Mondrian partition; c) Mean-squared error; d) Operational error.

## 3.2 Non-gridded data estimation

A fundamental advantage of the ESI model in comparison to traditional geostatistics is its capacity to effortlessly analyse d-dimensional[3], non-gridded data. As a result, the `spatialize` library is able to automatically generate spatial estimates for any set of points in space, even when these are not arranged on a regular grid. This includes irregularly-spaced points, incomplete grids, and 2D surfaces with variations on a third axis (frequently termed 2.5D). In this sense, it is highly flexible.

In this section, we employ the `esi_nongriddata()` function, which generates estimates from a set of sample points at a set of unmeasured points at arbitrary locations in space. Employing non-gridded sample data[4], we will present a comparison of the estimates derived from three different Kriging implementations and ESI-Kriging (i.e. using Universal Ordinary Kriging as base interpolator).

The dataset employed for this example, as well as the corresponding Ordinary Kriging estimates, are available to be loaded in the `spatialize` library, as shown in Code snippet 13.

---

**Code snippet 13** Loading the samples, locations, and ordinary kriging estimate for the `drill_holes_andes_2D` dataset in `spatialize`.

---

```
from spatialize.data import load_drill_holes_andes_2D

samples, locations, krig, _ = load_drill_holes_andes_2D()
```

---

This is a set of 400 copper grade data, placed non-regularly, with coordinates in the space of 60,000 points.

---

[3]Currently, spatialize offers support for up to two dimensions when using Voronoi partitions and five dimensions when using Mondrian. For the 4D (space-time) and 5D (spatial with two angles, for fault description, for example) case, the implementation includes only IDW as local ESI interpolator.

[4]In this case, the estimates will be calculated on a set of points arranged as a grid so that a comparison with Ordinary Kriging is possible. However, Spatialize could be implemented over any given set of points.



### 3.2.1 ESI vs Kriging

To produce the expert estimate using Ordinary Kriging in the example, an omnidirectional experimental variogram was generated, used and fitted to a function (theoretical variogram) containing two nested spherical structures (Equation 13).

$$\gamma(h) = \text{sill}_1 \left( 1.5 \frac{h}{\text{range}_1} - 0.5 \left( \frac{h}{\text{range}_1} \right)^3 \right) + \text{sill}_2 \left( 1.5 \frac{h}{\text{range}_2} - 0.5 \left( \frac{h}{\text{range}_2} \right)^3 \right) \tag{13}$$

Where:

$$\text{sill}_1 = 0.17, \quad \text{range}_1 = 95, \quad \text{sill}_2 = 0.14, \quad \text{range}_2 = 220$$

Both experimental and theoretical variograms are shown in Figure 16.

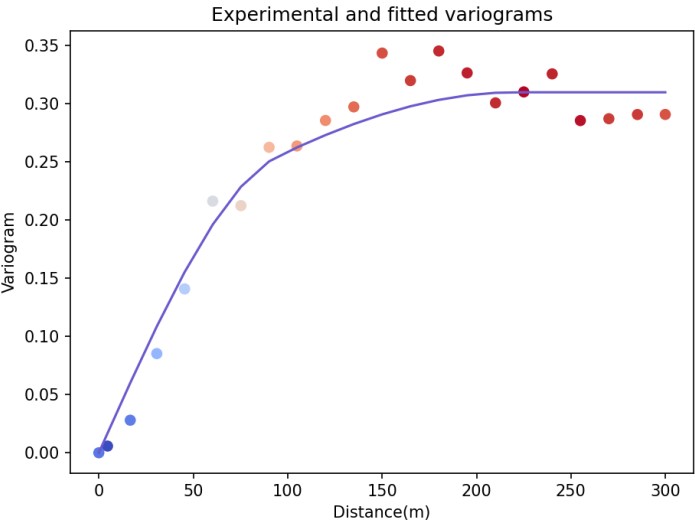

**Figure 16.** Experimental (dots) and theoretical (blue line) variograms for the Global Ordinary Kriging example.

Figure 17 (left) shows the data points (left), and the Global Ordinary Kriging estimation (right), which corresponds to the example developed in Egaña et al. (2021).

In addition, we have implemented an automated workflow that uses `scikit-learn` to run a parameter grid search, in order to obtain the best variogram model and fit the experimental variogram, and `PyKrige` to run Ordinary Kriging using

the best parameters found. Figure 18 shows the results for the automated implementation (right) alongside the manual expert implementation of Global Ordinary Kriging (left).

This automated approach differs from the process carried out by the expert in three ways: It uses global Kriging instead of domains depending on local stationarities; it determines variogram parameters in a heuristic way rather than through expert judgment; and it assumes that the variogram is the same in all directions (isotropy), ignoring potential anisotropies in the data.





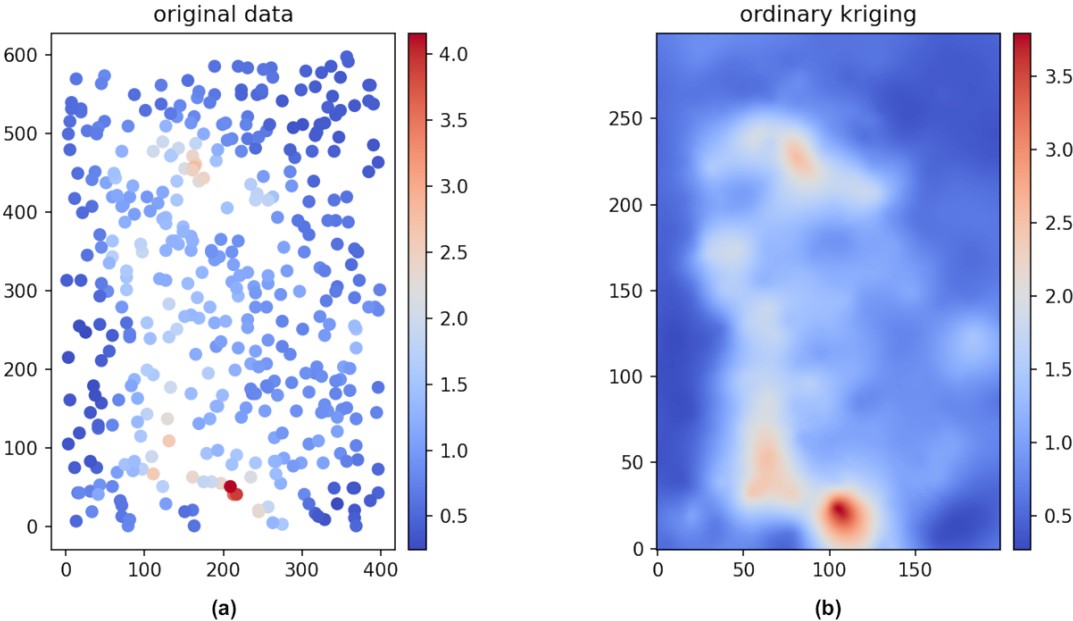

**Figure 17.** a) Map of points with values that are the input for the estimation in non-gridded data; b) Global Ordinary Kriging estimation based on those data points.

In this example, the result obtained for the parameter search is as follows:

```
best_score R² = 0.918
best_params = {'variogram_model': 'exponential'}
```

    As we can see in Figure 18, the estimation obtained with the automated algorithm is comparable to the ordinary Kriging performed manually on the basis of expert judgment. This is because the example is an ideal case of isotropy and stationarity, i.e. it fulfils the basic assumptions of a linear method such as kriging.

    Next, the same estimation previously performed with manual and automated Kriging was performed using the `spatialize`
library.

    First, the `esi_hparams_search()` function was employed to obtain the best parameters for the estimation. The evaluated set is shown in the following snippet:

```
search_result =
esi_hparams_search(points, values, xi,
local_interpolator="idw", griddata=False, k=10,
p_process="mondrian",
exponent=list(np.arange(1.0, 15.0, 1.0)),
```





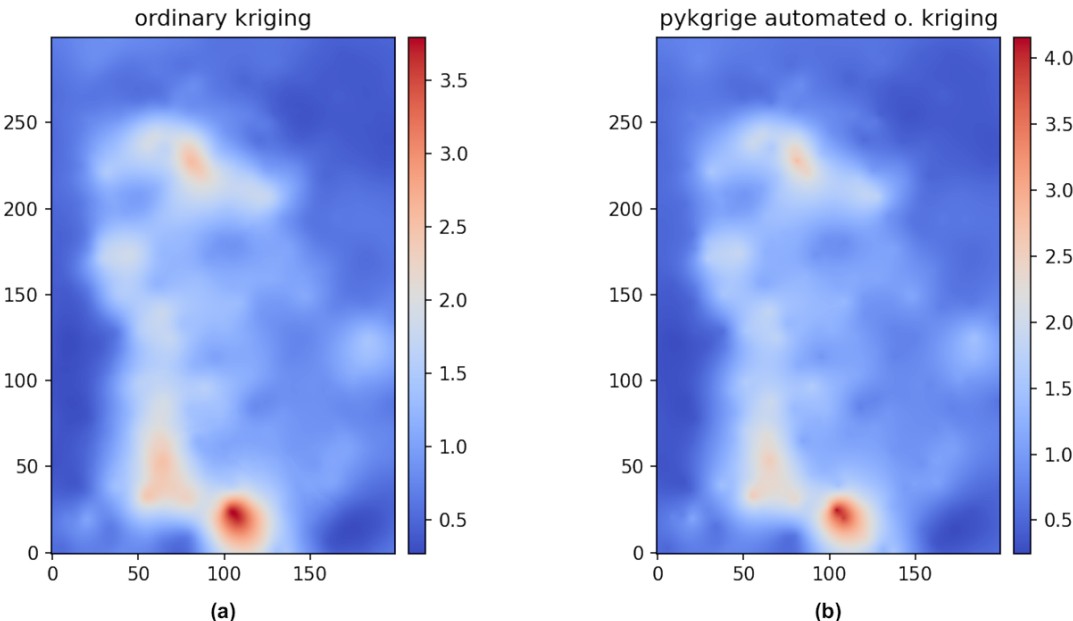

**Figure 18.** a) Global Ordinary Kriging estimation based on non-gridded example data points; b) Automated Kriging estimation with *pykrige*.

```
6              alpha=(0.5, 0.6, 0.8, 0.9, 0.95, 0.98)
7              )
```

Figure 19 shows the histogram for the cross-validation errors of the 168 search scenarios. In this case, the best-case scenario has the following parameters:

```
{'agg_func_name': 'mean', 'cv_error':
 0.10273662625968455, 'local_interpolator': 'idw', 'exponent': 8.0,
 'alpha': 0.9, 'result_data_index': 98, 'agg_function': <function mean
 at 0x14815f060>, 'p_process': 'mondrian'}
```

In this case, the local interpolator that ESI is using is IDW, and we can observe that the chosen exponent is relatively high. Recall that when the exponent is 0, the local estimate is the simple average among all neighbours, which generates a smoothing effect similar to Kriging. On the other hand, when the exponent tends to infinity, IDW becomes the nearest neighbour estimator, which implies that the estimate acquires a more abrupt change effect.

With these obtained parameters, we then performed the corresponding estimation using the `esi_nongriddata()` func-
tion, as shown in Code snipped 14.





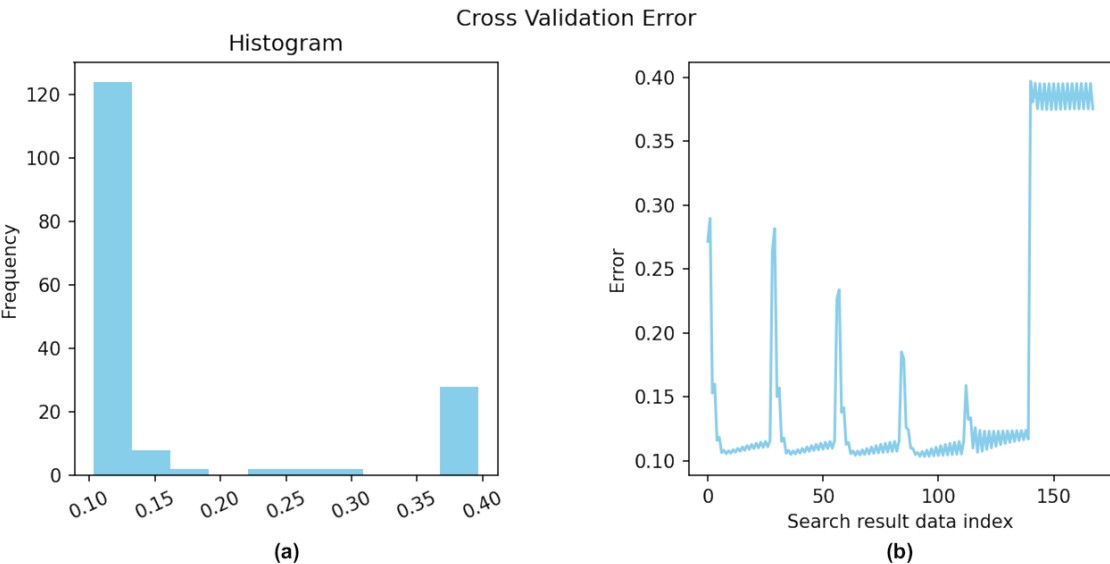

**Figure 19.** Cross validation error for the ESI-IDW non-gridded estimation parameter grid search. a) Histogram of errors; b) Errors in the sequence of scenarios during the search.

---

**Code snippet 14** Estimation with non-gridded data for ESI-IDW with the best parameters obtained from the previous grid search.

```
result =
esi_nongriddata(points, values, xi,
                local_interpolator="idw",
                p_process="mondrian",
                n_partitions=500,
                best_params_found=search_result.best_result()
                )
```

---

Once the prediction was been obtained, the `lf.OperationalErrorLoss()` function was used to generate a custom precision calculation. Specifically, a loss function called 'Operational Error' was defined as:

```
op_error = lf.OperationalErrorLoss(np.abs(np.nanmin(values) - np.nanmax(values)))
```

Figure 20 shows the resulting estimation and precision using the operational error loss function for the best case of the parameter grid for ESI-IDW. In this Figure, we can observe a more pixelated texture, which is consistent with what was
observed above regarding the effect of the abrupt change of a relatively high-value exponent for the local IDW interpolator. Interestingly, this contrasts with the perception, conditioned by the widespread use of Kriging, that a smoothed estimate may be better.




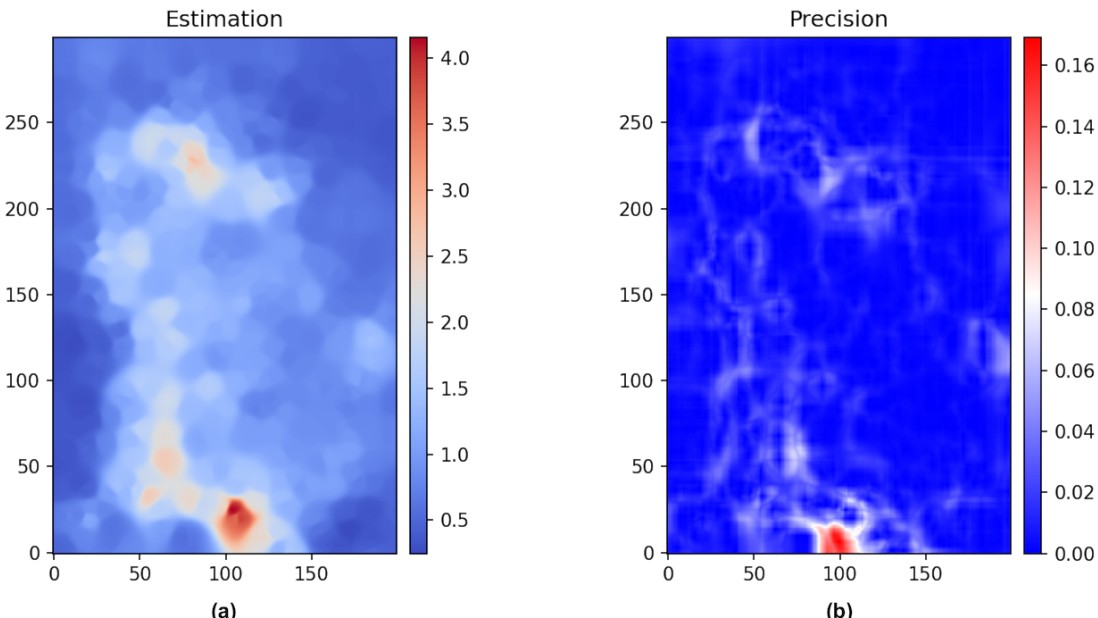

**Figure 20.** a) Best parameter non-gridded estimation with ESI-IDW; b) Precision obtained with operational error loss function.

Next, the process of searching for parameters and then estimating using the best set was repeated, but Voronoi was used as the partitioning method.

Figure 21 shows the histogram for the cross-validation errors of the 336 search scenarios. Note that using the Voronoi partitioning method doubles the number of search scenarios due to the `data_cond` parameter, which takes two possible default values (to condition or not condition the partitioning to the samples). In this case, the best-case scenario has the following parameters:

```
{'agg_func_name': 'mean', 'cv_error':
0.10132136752665043, 'local_interpolator': 'idw', 'exponent': 8.0,
'data_cond': True, 'alpha': 0.5, 'result_data_index': 14,
'agg_function': <function mean at 0x16827b060>,
'p_process': 'voronoi'}
```

Note that in this case, as in the previous case of ESI-IDW parameter search with the Mondrian partitioning method, the
exponent for the local interpolator is relatively high. This implies that the estimation should appear to have abruptly changing textures.





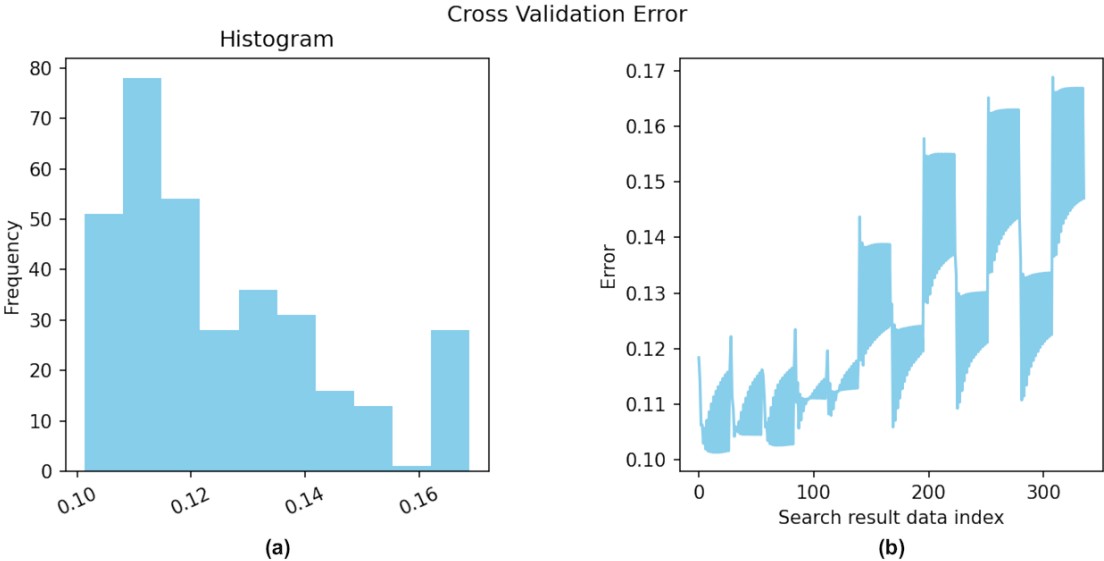

**Figure 21.** Cross validation error for the ESI-IDW non-gridded estimation parameter grid search, using Voronoi partition method. a) Histogram of errors; b) Errors in the sequence of scenarios during the search.

Next, the corresponding estimation, using the Voronoi partition method, was performed using the `esi_nongriddata()` function. Figure 22 shows the resulting estimation and precision using the operational error loss function for the best case of the parameter grid for ESI-IDW.

It can indeed be seen that, as in the case of the use of Mondrian as a partitioning method, the texture of the estimation is one of abrupt changes. On the other hand, the precision map is in a notoriously larger range of values, suggesting that, for the evaluated grid of parameters, the optimum for the case of the Voronoi partitioning method achieves a lower accuracy. In this sense, it is important to note that although the same search range was used for the parameter `alpha` in both examples, this parameter has a different implementation in each case – although it reflects the granularity of the partition in both methods –

which explains this difference. This argument is reinforced by the fact that the resulting value for this parameter (0.5) is at the lower end of the range used.





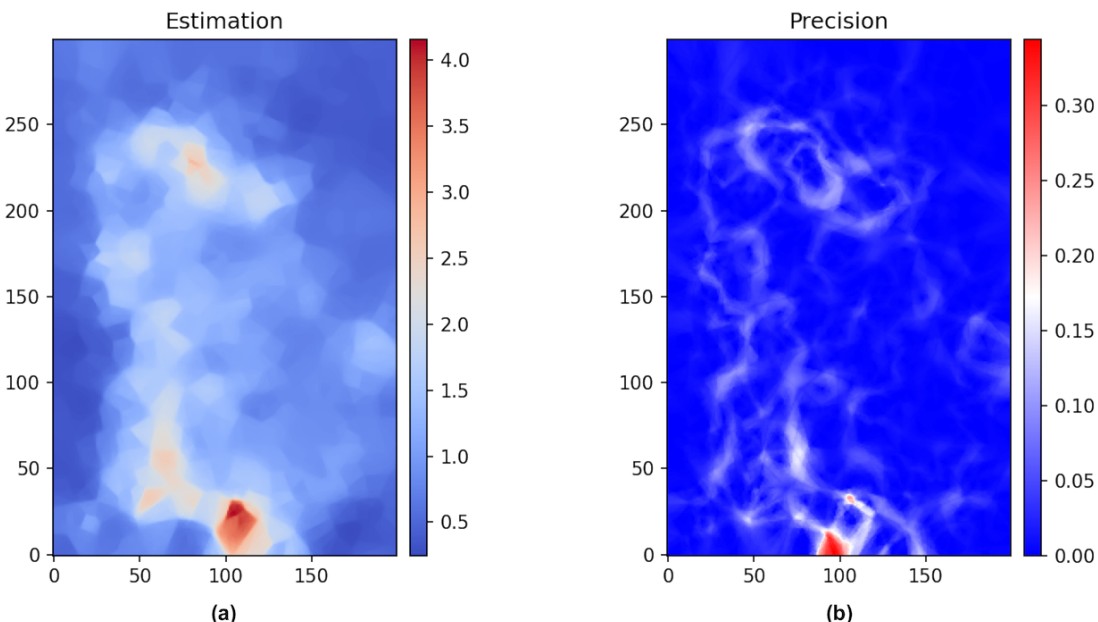

**Figure 22.** a) Best parameter non-gridded estimation with ESI-IDW; b) Precision obtained with operational error loss function, this time with Voronoi partition method.

Analogous to the ESI-IDW case, we present an estimation for the ESI-Kriging case (using Kriging as base interpolator for ESI)[5]. In this case, the parameter grid includes four models for the omnidirectional covariance function `model`:

```
search_result =
esi_hparams_search(points, values, xi,
                    local_interpolator="Kriging", griddata=False, k=10,
                    model=["spherical", "exponential", "cubic", "gaussian"],
                    nugget=[0.5, 1.0],
                    range=[100.0, 500.0, 1000.0],
                    alpha=list(np.flip(np.arange(0.90, 0.95, 0.01))),
                    sill=[0.9, 1.0, 1.1]
                    )
```

In Figure 23, we can see the cross-validation error for the scenarios generated with this parameter search.

---

[5]For the local Kriging interpolator, it is not possible to use the Voronoi partitioning method; this option has not yet been implemented in `spatialize`



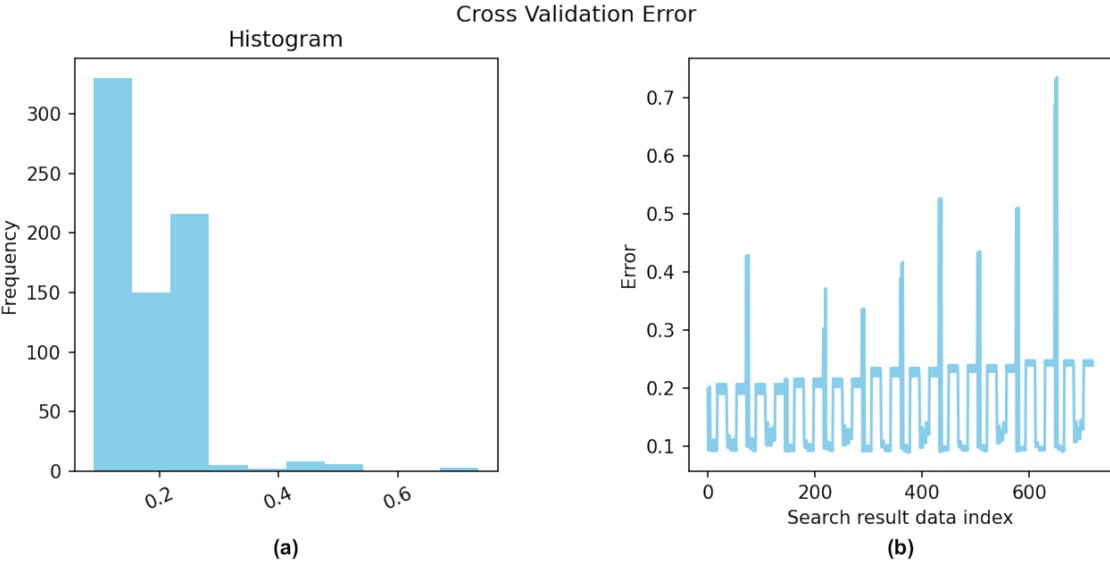

**Figure 23.** Cross-validation error for the ESI-Kriging non-gridded estimation parameter grid search. a) Histogram of errors; b) Errors in the sequence of scenarios during the search.

Once the best set of parameters was obtained, we generated the corresponding estimates using the `esi_nongriddata()` function. In addition, we generated an accuracy estimate using the operational error in the same way as in the ESI-IDW case. The calculated estimate and precision are shown in Figure 24.




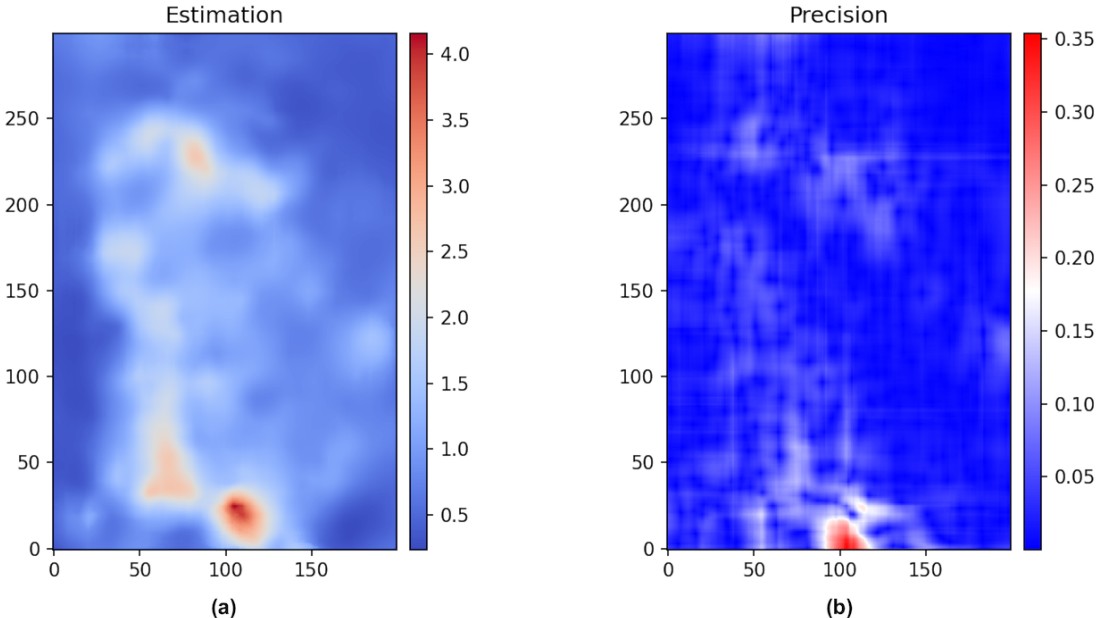

**Figure 24.** a) Best parameter non-gridded estimation with ESI-Kriging; b) Precision obtained with operational error loss function.

Notably, in the ESI-Kriging example in Figure 24 less smoothing can be noted compared to the Kriging example (Figure 18), which is generally more credible as an interpolation result.

### 3.2.2  3D non-gridded data estimation

As a complementary analysis, to demonstrate ESI's generalisation capability, the output for a three-dimensional non-gridded data estimation is presented in Figure 25. The data corresponds to that loaded by using the `load_drill_holes_andes_3D()` function in `spatialize.data`.





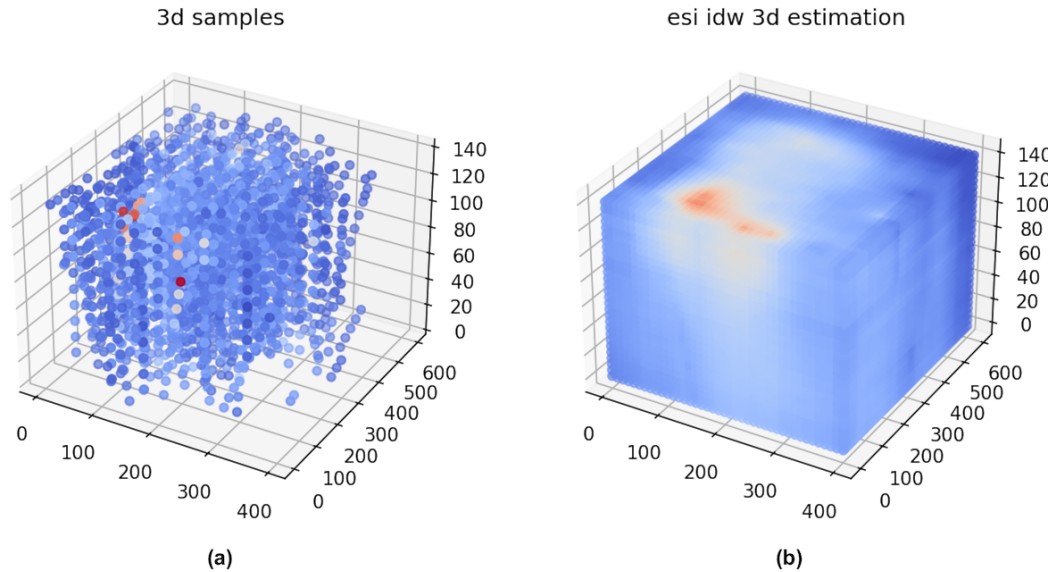

**Figure 25.** a) 3D data samples available; b) Resulted estimation using ESI-IDW.

## 4   Conclusion and future work

We have introduced `spatialize`, an open-source library that makes available to the community an efficient implementation
of a highly novel geostatistical technique. It aims to provide a general-purpose tool for non-expert geostatistics users that
provides automated tools that are on par with expert-use geostatistical tools.

The implemented technique is known as *ensemble spatial interpolation* (ESI). It is essentially data-driven, its philosophy
is aligned with computational statistics, and it seeks to bring the power of ensemble learning to geostatistical practice. In this
sense, the main strength of `spatialize` is that it requires minimal user intervention, and in cases where it is necessary (such
as in the choice of some hyperparameters), it provides help tools that facilitate the process.

The idea is that `spatialize` will be, in the medium term, one of the best open-source geostatistical libraries available in
the Python language. To this end, we have a roadmap that includes, as future work:

– Add other partition generation processes, such as Mondrian processes with random rotations, to allow for more expres-
siveness in the set of generated partitions.

– Add local interpolators that allow local adaptability in the calculation of ESI samples.

– Add local interpolators that allow the use of spatial statistical tools, such as CAR-type models.

– Add support for estimation of categorical variables.



- On the computational side, add support for the use of GPUs to allow integration with tools such as Google Colab.

- Add classical geostatistical functionalities for expert users who require them. This includes a more general Kriging implementation, allowing nested structures, for example.

*Code and data availability.* Both the source code for `spatialize` and the usage examples shown above are available in the spatialize project on Github (Navarro et al., 2025b), which can be accessed at https://github.com/alges/spatialize. The version presented in this paper is archived on Zenodo at https://doi.org/10.5281/zenodo.16782612 (Navarro et al., 2025a), along with a supplementary User Manual.
In particular, the usage examples are available to be run in the `examples/scripted_examples` folder.

The gridded data estimation example (Section 3.1) employs the following scripts:

1. `esi_griddata.py`
2. `esi_griddata_mondrian_voronoi.py`
3. `esi_grid_search.py`
4. `idw_grid_search.py`
5. `idw_griddata.py`
6. `esi_custom_precision.py`

The non-gridded data estimation example (Section 3.2) employs the following scripts:

1. `esi_nongriddata.py`
2. `pykrige_example.py`
3. `esi_3d_nongriddata.py`

The data that is used in the scripted examples is part of the `spatialize` library package and is installed with it in `spatialize.resources`. Besides, the `spatialize.data` module contains the functions for loading this data.

*Author contributions.* Research conceptualisation, paper preparation, and analysis were performed by AFE, AE and FN. The ESI framework was first developed by FG, FN and AFE. FN, AFE, AE, FG, MJV, and JFS contributed to the paper edits and technical review.

*Competing interests.* The contact author has declared that none of the authors have any competing interests.

*Acknowledgements.* This work was supported in part by the Chilean National Agency for Research and Development (ANID) under Project FONDEF IT23I013, FONDEF TA24I10053 and ANID/BASAL AFB230001. We acknowledge the computational resources provided, the
useful insights, and the support provided by the whole ALGES team.





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
