# Peer review of "Spatialize v1.0: A Python/C++ Library for Ensemble Spatial Interpolation"

_EGUsphere, 2025_

## Referee Comment (RC1)

**Referee Report**

Title: Spatialize v1.0: A Python/C++ Library for Ensemble Spatial Interpolation

Author(s): Felipe Navarro et al.

MS No.: egusphere-2025-3272

MS type: Development and technical paper

The authors present a Python library, `Spatialize`, which implements several spatial interpolation methods with automated hyperparameter calibration. The paper addresses an important need in the geosciences community for accessible spatial interpolation tools. However, the manuscript requires substantial revisions before it can be considered for publication.

The authors claim that the package is designed for experts and non-experts with minimal geostatistical knowledge. However, as an economist with an interest in climate data, I think the implementation still requires a fair level of understanding of the underlying model and basic parameters especially if you plan to do parameter calibration. An initialization is required for the library to conduct a grid search.

In addition, the paper does not clearly articulate what `Spatialize` can do that existing libraries (`SciPy`, `PyKrige`, and `scikit-learn`) cannot. The authors should clearly state which capabilities are unique to `Spatialize`, a table of performance comparison would be appreciated.

**Major comments:**

The flow of the paper is chaotic and fragmented. The authors present a series of simulation and validation, but they lack a coherent framework of how the examples are related, or build upon each other.

The performance evaluation relies mostly (if not solely) on graphical presentations, lacking numerical support. When performance is similar, it is difficult to identify the differences between figures, such as Figures 20 and 22. A table of quantitative metrics should be presented.

The validation is solely based on simulation data. A real world application would help a lot for demonstrating how the library can be applied in empirical studies.

The library supports high dimension interpolation, such as space-time variation, this is theoretically interesting as it can capture the dynamic special dependencies if they exist. But if this makes sense in practice remains unknow. If high-dimensional interpolation is

a key feature of the library, a real-world example demonstrating its necessity and showing how the library improves performance would be helpful.

It is not clear how ensembling multiple models outperforms the predictions of a single model, nor how the ensembling function is defined.

**Specific comments:**

Given that the stated target users include non-experts, it would be helpful to provide intuitive explanations of what each algorithm does in the algorithm descriptions.

In line 11, the period before the parenthesis citation should be removed. "...point locations. (Li and Heap, 2014)." should be "... point locations (Li and Heap, 2014)." The same applies to line 77.

Figures are not sufficiently discussed. For example, Figure 8 (a) is only mentioned in terms of the name, no explanation why the errors are clustered in low and high levels, but fewer observations have middle level errors. Also according to Figure 8 (b), it seems index 600 is lower than index 302, contrary to line 284, which states that the lowest error is located at index 302?

The function in Code snippet 1 has wrong indentation. Line 2 should be indented.

---

## Referee Comment (RC2)

**General comments:**

The manuscript "Spatialize v1.0: A Python/C++ Library for Ensemble Spatial Interpolation" introduces a python package "`spatialize`". The methodology is based on a previous publication (Egaña et al., 2021). The motivation of "`spatialize`" is to provide geostatistical tools to non-experts that lack the experience of spatial analysis, i.e., regarding spatial interpolation. The implemented ESI approach replaces the expert knowledge of a modeler with an ensemble based estimation and grid search for hyperparameters.

In my view, the manuscript lacks the necessary clarity in its comparison analysis. Comparisons are only carried out through a visual assessment of results and precision maps. Typical statistics like RMSE, MAE and alike are missing. Furthermore, to underline the added value of simplifying the application for non-experts lacks a code based comparison with existing python implementations for spatial interpolation. It would also be interesting to see in this manuscript how robust the approach is, i..e how well miss-specifications can be compensated. Targeting at non-experts, a clear road map with guidance and caveats would also be beneficial.

**Specific comments:**

- classical approaches are not limited to gridded data, neither kriging nor IDW (lines 328/329)
- Figures with grid search results could benefit from indication which parameters are currently investigated; the jig-saw pattern might e.g., be due to different variogram types
- how is the sill obtained? The code snippets only list range and nugget as parameters
- does/can the grid search also optimize the data splits, i..e tree configurations?
- The 3D case only shows possibilities, but lacks any explanation or discussion appropriate for a manuscript (in contrast to, e.g., a manual)
- comparisions lacks a number based comparisions MSE/RMSE/MAE and a like
- In case of the simple mean aggregation and IDW with p=1, is there an actual benefit of ESI? To my understanding, and under the assumption than on average all tree induced partitions would have the same sum of distances of its members to x*, The ESI approach would just be IDW with more points.

**Technical corrections:**

- Package name is typeset in different format, recommendation to set it always as fixed width font.
- Typo: line 179 "...ion 9rep…"

---

## Author Comment (AC1)

Spatialize v1.0: A Python/C++ Library for Ensemble Spatial Interpolation

**Response to reviewers**

Dear reviewer,

We sincerely appreciate the time and effort that you have dedicated to providing valuable feedback on our manuscript. We are truly grateful for your insightful comments on our paper. We find them valuable and constructive. We have provided a point-by-point response to your comments and concerns (in blue). Additionally, certain figures have been adjusted, and we have prepared a new manuscript that incorporates all the changes.

**Referee #1**

*The authors present a Python library, Spatialize, which implements several spatial interpolation methods with automated hyperparameter calibration. The paper addresses an important need in the geosciences community for accessible spatial interpolation tools. However, the manuscript requires substantial revisions before it can be considered for publication.*

*The authors claim that the package is designed for experts and non-experts with minimal geostatistical knowledge. However, as an economist with an interest in climate data, I think the implementation still requires a fair level of understanding of the underlying model and basic parameters especially if you plan to do parameter calibration. An initialization is required for the library to conduct a grid search.*

*In addition, the paper does not clearly articulate what Spatialize can do that existing libraries (SciPy, PyKrige, and scikit-learn) cannot. The authors should clearly state which capabilities are unique to Spatialize, a table of performance comparison would be appreciated.*

This is an important observation. The manuscript has been revised to clearly explain the underlying model and calibrate the parameters. We have reorganized the manuscript to better introduce each concept, facilitating an easy transition from simple to complex examples using real data.

Additionally, a new section has been incorporated under the title of "The Spatialize Library". The purpose of this section is to provide a concise introduction to the library, whilst also comparing it with existing libraries through a table of performance comparison (**Table 1**).

**Major comments:**
*The flow of the paper is chaotic and fragmented. The authors present a series of simulation and validation, but they lack a coherent framework of how the examples are related, or build upon each other.*

We hope that incorporating the "The Spatialize Library" section results in a more seamless progression between the description of the ESI algorithm and the usage examples.

Moreover, the examples have been restructured into two distinct "case studies" to accurately reflect the intended manner of use of the library by its users, rather than the previous separation between gridded and non-gridded implementations with arbitrary examples.

*The performance evaluation relies mostly (if not solely) on graphical presentations, lacking numerical support. When performance is similar, it is difficult to identify the differences between figures, such as Figures I/ and II. A table of quantitative metrics should be presented.*

We have incorporated tables with MAE, RMSE and MSE metrics in order to provide an explicit numerical performance evaluation. However, for the real-world example (copper grade dataset), we are only able to offer cross-validation metrics (see explanation in the next point).

*The validation is solely based on simulation data. A real world application would help a lot for demonstrating how the library can be applied in empirical studies.*

The copper grade drill holes dataset employed for the non-gridded example corresponds to a real-world application. We understand that our previous manuscript was not very clear in this aspect, which is why a thorough description of the datasets has been added (Section **4.1**). We expect that this is now clearer.

The choice to use a synthetic dataset –besides the real-world drill hole dataset– is because synthetic scenarios allow for better performance evaluation: in real-world applications, reference maps are not usually available, since measurements are taken at specific sampling locations. The decision to use simulation data is made so that performance metrics can be calculated across unmeasured locations. Due to the sparse nature of real-world data, we are only able to provide numerical evaluations for locations with available measurements and cross-validation methods.

*The library supports high dimension interpolation, such as space-time variation, this is theoretically interesting as it can capture the dynamic special dependencies if they exist. But if this makes sense in practice remains unknow. If high-dimensional interpolation is a key feature of the library, a real-world example demonstrating its necessity and showing how the library improves performance would be helpful.*

While high-dimensional interpolation is indeed a distinctive feature of the library, we have intentionally omitted such examples for two key reasons:

First, the primary objective of our case studies is to demonstrate the practical usage of the library's various tools, enabling readers to integrate them into their own analyses. Since the

syntax remains consistent regardless of dimensionality, a high-dimensional example would not provide additional methodological insight beyond what is already presented.

Second, a comprehensive real-world spatio-temporal dataset would necessitate substantially more complex visualizations and extensive analysis—requiring a separate publication to do it justice. We have prioritized simpler, more focused applications to clearly illustrate each tool's functionality while maintaining a reasonable manuscript length.

*It is not clear how ensembling multiple models outperforms the predictions of a single model, nor how the ensembling function is defined.*

ESI's ensemble approach outperforms single-model predictions by combining multiple local perspectives and reducing sensitivity to individual partition configurations.

In traditional interpolation, a single model uses all available data but applies uniform assumptions across the entire domain. This can lead to poor predictions in regions where local spatial structures differ from global patterns. ESI addresses this by creating multiple local estimates, each based on a random partition. While individual local estimates may be unstable (since they use fewer samples and depend on the specific partition configuration), aggregating many of them stabilizes the predictions while preserving sensitivity to local spatial patterns.

Specifically, the random partitioning and aggregation ensure that:
-   Samples closer to the target location appear together in partition cells more frequently, naturally receiving higher effective weight in the final estimate without requiring explicit distance calculations or neighborhood definitions.
-   Each partition captures different local spatial configurations. Aggregating across partitions averages out errors or biases from any single partition while reinforcing consistent local patterns that appear across multiple partitions.
-   The resulting distribution of estimates enables uncertainty quantification, which single-model approaches cannot provide.

The ensemble function is simply the mean, median, or mode of the estimates across all partitions for each target point.

**Specific comments:**
*Given that the stated target users include non-experts, it would be helpful to provide intuitive explanations of what each algorithm does in the algorithm descriptions.*

Algorithm 1 is explained in lines **104-111**. If this explanation does not address your concern, we would appreciate further clarification on what additional information would be helpful.

*In line 11, the period before the parenthesis citation should be removed. "...point locations. (Li and Heap, 2014)." should be "... point locations (Li and Heap, 2014)." The same applies to line 77.*

Thank you, this issue has been solved (lines **11** and **77**).

*Figures are not sufficiently discussed. For example, Figure 8 (a) is only mentioned in terms of the name, no explanation why the errors are clustered in low and high levels, but fewer observations have middle level errors. Also according to Figure 8 (b), it seems index 600 is lower than index 302, contrary to line 284, which states that the lowest error is located at index 302?*

This issue has been addressed in the revised manuscript through the clarification of examples and the reorganisation of the text. Additionally, we have enhanced the figure discussion and replaced the figures with colour-blind-compatible versions. The mentioned indexing issue has been resolved.

*The function in Code snippet 1 has wrong indentation. Line 2 should be indented.*

This issue has been solved (around line **274)**.

---

## Author Comment (AC2)

**Response to reviewers**

Dear reviewer,

We sincerely appreciate the time and effort that you have dedicated to providing valuable feedback on our manuscript. We are truly grateful for your insightful comments on our paper. We find them valuable and constructive. We have provided a point-by-point response to your comments and concerns (in blue). Additionally, certain figures have been adjusted, and we have prepared a new manuscript that incorporates all the changes.

**Referee #2**

*The manuscript "Spatialize v1.0: A Python/C++ Library for Ensemble Spatial Interpolation" introduces a python package "spatialize". The methodology is based on a previous publication (Egaña et al., 2021). The motivation of "spatialize" is to provide geostatistical tools to non-experts that lack the experience of spatial analysis, i.e., regarding spatial interpolation. The implemented ESI approach replaces the expert knowledge of a modeler with an ensemble based estimation and grid search for hyperparameters. In my view, the manuscript lacks the necessary clarity in its comparison analysis. Comparisons are only carried out through a visual assessment of results and precision maps. Typical statistics like RMSE, MAE and alike are missing. Furthermore, to underline the added value of simplifying*
*the application for non-experts lacks a code based comparison with existing python implementations for spatial interpolation. It would also be interesting to see in this manuscript how robust the approach is, i..e how well miss-specifications can be compensated. Targeting at non-experts, a clear road map with guidance and caveats would also be beneficial.*

This comment has been addressed by incorporating additional information. The revised text has incorporated tables with MAE and RMSE metrics in order to provide a more explicit numerical performance evaluation.

This *added value of simplicity* is not dependent on the code implementation; rather, it is related to the preliminary work conducted by kriging practitioners to perform spatial modelling, a requirement prior to implementing kriging. This process typically involves the use of supplementary tools and the application of expert knowledge to determine one or more variogram models, which must be manually specified when using kriging.

Libraries such as PyKrige provide support for the use of the grid search utility of scikit-learn in order to determine the optimal variogram model, although only basic variogram models are incorporated. More complex spatial structures require the use of advanced tools for variogram modelling.

When using ESI with kriging as a local interpolator, while it is true that a variogram model must be specified, the robustness of the kriging implementation becomes less important given the ensemble scheme. There are two key reasons for this. (a) In situations involving complex spatial structures that may require a complex variogram, a simple variogram will likely suffice on a local scale once the space has been partitioned; and (b) the whole point of ensemble models is that less robust individual models can be used, since we are combining the results of many "weak" models.

We understand that adding a misspecification could demonstrate the robustness of our ensemble modelling approach. This is a reasonable observation from the perspective of traditional modelling approaches. However, as mentioned previously, since we are combining the results of many 'weak' models, this misspecification will directly affect the outcome. For example, if you add an incorrect nugget or range when using the kriging interpolator, the grid search will not be able to find optimal results.

We also appreciate the suggestion to enhance accessibility for non-expert users. First, we have added "The Spatialize Library" section, which provides an introduction to the library and its main features. Users can consult the documentation for more detailed information on specific parameters and advanced functionality. Second, we have restructured the usage examples into case studies that follow a standard procedure for implementing ESI. These case studies demonstrate both the core workflow and optional features or variants, providing readers with a practical roadmap for applying the library to their own analyses.

Additionally, we will soon publish a series of complementary tutorials and examples, along with complete documentation in Read the Docs format, to further support novice users.

**Specific comments:**
*Classical approaches are not limited to gridded data, neither kriging nor IDW (lines 328/329)*

Thank you for this important clarification. You are correct that libraries like PyKrige, SciPy, and GSTools do offer interpolation for non-gridded data. We have revised the manuscript to accurately reflect this capability.

The key advantages of Spatialize lie in its flexibility and scope. Specifically, Spatialize supports interpolation up to 5 dimensions (not available in PyKrige or SciPy), provides multiple user-friendly interpolation methods with accessible interfaces (unlike GSTools, which focuses exclusively on kriging with user-specified parameters), and offers comprehensive tools, including parameter search functions for both gridded and non-gridded implementations.

We have updated Table 1 to clearly present these distinctions and provide a more accurate comparison across libraries.

*Figures with grid search results could benefit from indication which parameters are currently investigated; the jig-saw pattern might e.g., be due to different variogram types*

Thank you for this suggestion. We have revised all grid search implementations to clearly specify the parameter configurations being employed in each case.

*How is the sill obtained? The code snippets only list range and nugget as parameters*

When not explicitly specified in the grid search or ESI implementation, the sill parameter defaults to a value of 1.

*Does/can the grid search also optimize the data splits, i..e tree configurations?*

The partitions themselves are generated randomly and cannot be directly controlled. However, the grid search can optimize several aspects of the partitioning process, including the number of partitions and their coarseness. Additionally, users can select the partitioning algorithm (Mondrian or Voronoi) and set a seed for reproducibility.

*The 3D case only shows possibilities, but lacks any explanation or discussion appropriate for a manuscript (in contrast to, e.g., a manual)*

We appreciate this feedback and have removed the 3D example. The primary objective of our case studies is to demonstrate the practical usage of the library's tools with sufficient detail and analysis. Since the syntax remains consistent regardless of dimensionality, we have focused on two 2-dimensional examples that allow for clearer visualization and more thorough discussion of the results and methodology.

*Comparisions lacks a number based comparisions MSE/RMSE/MAE and alike*

Thank you for this suggestion. We have incorporated tables with MAE, RMSE, and MSE metrics to provide quantitative performance evaluation. For the synthetic data case study, these metrics are calculated across the entire estimation grid. For the real-world example, we use both leave-one-out cross-validation and k-fold cross-validation.

*In case of the simple mean aggregation and IDW with p=1, is there an actual benefit of ESI? To my understanding, and under the assumption than on average all tree induced partitions would have the same sum of distances of its members to x\*, The ESI approach would just be IDW with more points.*

Even with p=1 and mean aggregation, ESI differs fundamentally from plain IDW due to its use of local interpolation, which considers only the samples within the vicinity of the target location.

In plain IDW, all sample points contribute with weights inversely proportional to distance, meaning distant points have a small but non-zero influence. In contrast, ESI creates a different local neighborhood in each partition by restricting estimation to only the samples within the same partition cell as the target location. When aggregating across partitions, ESI computes the

mean of ratios rather than a single ratio of sums. Consequently, samples closer to the target appear in the same cell more frequently, gaining higher effective weight, while distant samples are excluded from most partitions entirely. This creates a probabilistic soft neighborhood that emphasizes local spatial structures, which is advantageous when studying spatially heterogeneous phenomena. In contrast, traditional IDW captures both global and local structures simultaneously.

**Technical corrections:**
*Package name is typeset in different format, recommendation to set it always as fixed width font.*

Thank you for this observation. We have revised the manuscript to consistently typeset all instances of the package name in fixed-width font using a verbatim environment.

*Typo: line 179 "...ion 9rep..."*

This issue has been solved (line **179**).